# Nucleoporin 153 links nuclear pore complex to chromatin architecture by mediating CTCF and cohesin binding

Shinichi Kadota[1,2,3], Jianhong Ou [1,3], Yuming Shi[1,2,3], Jeannie T. Lee [4,5], Jiayu Sun [1,2,3] & Eda Yildirim [1,2,3✉]

Nucleoporin proteins (Nups) have been proposed to mediate spatial and temporal chromatin organization during gene regulation. Nevertheless, the molecular mechanisms in mammalian cells are not well understood. Here, we report that Nucleoporin 153 (NUP153) interacts with the chromatin architectural proteins, CTCF and cohesin, and mediates their binding across *cis*-regulatory elements and TAD boundaries in mouse embryonic stem (ES) cells. NUP153 depletion results in altered CTCF and cohesin binding and differential gene expression — specifically at the bivalent developmental genes. To investigate the molecular mechanism, we utilize epidermal growth factor (EGF)-inducible immediate early genes (IEGs). We find that NUP153 controls CTCF and cohesin binding at the *cis*-regulatory elements and POL II pausing during the basal state. Furthermore, efficient IEG transcription relies on NUP153. We propose that NUP153 links the nuclear pore complex (NPC) to chromatin architecture allowing genes that are poised to respond rapidly to developmental cues to be properly modulated.

[1] Department of Cell Biology, Duke Medical Center, Durham, NC 27710, USA. [2] Duke Cancer Institute, Duke University, Durham, NC 27710, USA. [3] Regeneration Next, Duke University, Durham, NC 27710, USA. [4] Department of Molecular Biology, Massachusetts General Hospital, Boston, MA 02114, USA. [5] Department of Genetics, The Blavatnik Institute, Harvard Medical School, Boston, MA 02114, USA. ✉email: eda.yildirim@duke.edu

Establishment of cell lineage specification, maintenance of cellular states, and cellular responses to developmental cues rely on gene regulation and spatial genome organization during development[1,2]. Emerging data point to highly coordinated activity between epigenetic mechanisms that involve nuclear architecture, chromatin structure, and chromatin organization[3,4]. However, our understanding on how nuclear architectural proteins are causally linked to chromatin organization and impact gene regulation have been limited underscoring the importance of defining the molecular determinants.

Nuclear architecture is in part organized by the nuclear lamina composed of lamin proteins and the nuclear pore complex (NPC). Nucleoporin proteins (Nups) are the building blocks of the NPC, which forms a ~60–120 mega dalton (mDa) macromolecular channel at the nuclear envelope mediating nucleocytoplasmic trafficking of proteins and RNA molecules during key cellular processes such as cell signal transduction and cell growth[5]. Beyond its role in nuclear transport, the NPC has been one of the nuclear structural sites of interest for its potential role in gene regulation by directly associating with genes[6]. Studies in budding yeast and metazoans have shown that the NPC provides a scaffold for chromatin modifying complexes and transcription factors, and mediates chromatin organization. In metazoans, such compartmentalization supports nucleoporin–chromatin interactions that influence transcription[7–9]. In yeast, inducible genes including *GAL*, *INO1*, and *HXK1* localize to the NPC upon transcription activation — a process that has been proposed to be critical for the establishment of transcription memory[10–12]. For several of these loci, NPC association facilitates chromatin looping between distal regulatory elements and promoters[13,14]. Similar mechanism applies to the developmentally regulated ecdysone responsive genes in *Drosophila melanogaster*. Upon activation, ecdysone responsive genes exhibit NUP98-mediated enhancer–promoter chromatin looping at the NPC[15]. Notably, NUP98 has been shown to interact with several chromatin architectural proteins, including the CCCTC-binding factor, CTCF. These findings collectively suggest that Nups can facilitate chromatin structure in a direct manner by regulating transcription and in an indirect manner whereby Nup-mediated gene regulation relies on architectural proteins. Nevertheless, the functional relevance of Nup–architectural protein interactions in transcription regulation and chromatin structure is not well understood.

Chromatin architectural proteins, CTCF and the cohesin, facilitate interactions between *cis*-regulatory elements[16,17]. These interactions influence the formation and maintenance of long-range chromatin loops that underlie higher-order chromatin organization[18,19]. Long-range loops of preferential chromatin interactions, referred to as "topologically associating domains" (TADs), are stable, conserved across the species, and exhibit dynamicity during development[17,20]. Importantly, TADs segregate into transcriptionally distinct sub-compartments[21,22] and exhibit spatial positioning[23]. Current models argue that lamina–chromatin interactions may provide sequestration of specific loci inside the peripheral heterochromatin and promote the formation of a silent nuclear compartment[24,25]. Despite the close interaction between the nuclear lamina and the NPC, we still know very little on how NPC–chromatin interactions influence transcription and chromatin organization at the nuclear periphery.

In mammals, Nups show variable expression across different cell types and their chromatin binding has been attributed to cell-type-specific gene expression programs[6]. NUP153 is among the chromatin-binding Nups which have been proposed to impact transcription programs that associate with pluripotency and self-renewal of mammalian stem cells[26–28]. NUP153 binding sites have been detected at the promoters, across gene bodies, and enhancers[26–28]. Nevertheless, the molecular basis for how NUP153 association at the enhancers or promoters impact chromatin structure and transcription remain to be open questions.

Here, we directly test the relationship between NUP153–chromatin interactions and gene regulation in pluripotent mouse ES cells. Towards elucidating NUP153-mediated mechanisms of transcription, we further utilize immediate early genes (IEGs) at which transcription can be efficiently and transiently induced using growth hormones such as the epidermal growth factor (EGF) in HeLa cells[29]. We report that NUP153 interacts with cohesin and CTCF, and mediates their binding at enhancers, transcription start sites (TSS), and TAD boundaries in mouse ES cells. NUP153 depletion results in differential gene expression that is most prevalent at bivalent genes[4]. At the IEGs, NUP153 binding at the *cis*-regulatory elements is critical for CTCF and cohesin binding and subsequent POL II pausing. This function of NUP153 is essential for efficient transcription initiation of IEGs. Notably, IEGs exhibit a NUP153-dependent positioning to the nuclear periphery during the basal state and reposition even closer to the periphery upon transcriptional activation. Our findings reveal that IEG–NUP153 contacts are essential for IEG transcription via the establishment of a chromatin structure that is permissive for POL II pausing at the basal state. We propose that NUP153 is a key regulator of chromatin structure by mediating binding of CTCF and cohesin at *cis*-regulatory elements and TAD boundaries in mammalian cells. Through this function, NUP153 links NPCs to chromatin architecture allowing developmental genes and IEGs that are poised to respond rapidly to developmental cues to be properly modulated.

## Results

**Identification of CTCF and cohesin as NUP153 interacting proteins**. To understand the functional relevance of NUP153 in transcriptional regulation and chromatin structure, we utilized an unbiased proteomics screen using mouse NUP153 as bait in an affinity purification assay. We expressed FLAG-tagged mouse NUP153 (FLAG-mNUP153) in HEK293T cells and carried out immunoprecipitation (IP) followed by mass spectrometry (MS) (Fig. 1a and Supplementary Fig. 1a, b). We identified several known NUP153 interacting proteins including TPR[30], NXF1 (ref. [31]), SENP1 (ref. [32]), and RAN[33]. In addition, IP–MS revealed that NUP153 interacts with chromatin interacting proteins including the cohesin complex components, SMC1A, SMC3, and RAD21.

NUP153 has been mapped to enhancers and promoters in mammalian cells and has been implicated in transcription regulation[26–28]. Nevertheless, the mechanisms are not well understood. We, thus focused on the NUP153–cohesin interactions as cohesin mediates higher-order chromatin organization, and regulates gene expression by facilitating and stabilizing enhancer–promoter interactions together with CTCF[34]. We performed FLAG-NUP153 IP followed by western blotting and determined that NUP153 interacts with CTCF and cohesin subunits (Fig. 1b). To define the nuclear fraction at which NUP153 spatially interacts with CTCF and cohesin, we performed biochemical chromatin fractionation assay using HeLa cells as previously described[35] (Fig. 1c). Micrococcal nuclease (MNase) treatment of the nuclear fraction (P1) resulted in the elution of chromatin binding proteins into the soluble nuclear fraction (S3) (Fig. 1c, d). We detected NUP62, in the insoluble nuclear fraction (P2, +/− MNase) suggesting that the P2 contains the intact nuclear membrane including the nuclear envelope and

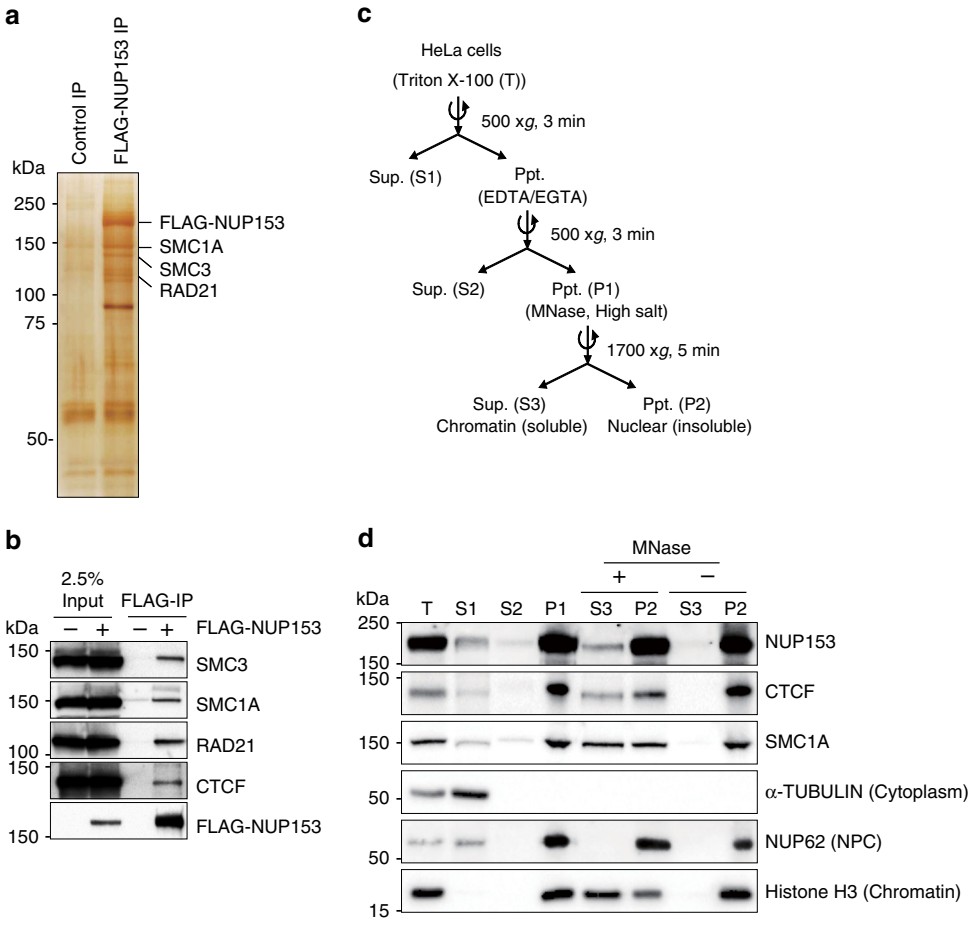

**Fig. 1 NUP153 interacts with CTCF and cohesin. a** Silver stain showing proteins that immunoprecipitated (IP) with FLAG-NUP153. **b** Co-IP shows FLAG-NUP153 interaction with CTCF, and cohesin subunits, SMC3, SMC1A, and RAD21. NUP153 was pulled down using anti-FLAG antibody. **c** Schematic showing steps of chromatin fractionation assay in HeLa cells. **d** NUP153 was detected in the nuclear insoluble fraction (P2) along with CTCF and cohesin. NUP153 detected in the chromatin-associated soluble fraction (S3) following micrococcal nuclease (MNase) treatment of P1 fraction. Ppt, precipitate; Sup, supernatant; Nucleoporin 62, NUP62; loading controls: α-TUBULIN (cytoplasm), Histone H3 (chromatin). Experiments were repeated twice. Source data are provided as a Source Data file.

the NPC. In accordance with earlier cell biological reports[36], we detected NUP153 both in the insoluble (P2), and the soluble (S3) nuclear fractions (Fig. 1d). This data suggested that the NUP153–chromatin interactions might be established at the nuclear periphery or in the nucleoplasm. Similar to NUP153, we detected a proportion of CTCF and cohesin in the insoluble nuclear fraction (P2) even in the presence of MNase (Fig. 1d). Insoluble fraction has been shown to contain nuclear matrix-associated proteins, including CTCF[37]. These findings argue that NUP153 may interact with CTCF and cohesin at the nuclear periphery, nuclear matrix, or within the nucleoplasm.

**NUP153 enrichment at the *cis*-regulatory elements and TAD boundaries**. NUP153 mediates transcription regulation of developmental genes in mouse ES cells[26]. Such function has been attributed to the transcriptional silencing role of NUP153 together with the Polycomb Repressive Complex 1 (PRC1). Nevertheless, only ~10% of NUP153 binding sites overlap with PRC1 interaction sites explaining only a small proportion of NUP153-mediated gene regulation in pluripotent mouse ES cells. To define NUP153-mediated chromatin structure and gene regulation, we mapped NUP153 binding sites using female mouse ES cell lines (EL16.7)[38] by DamID-Seq[39] (Supplementary Fig. 1c–e). A Dam only expressing cell line was used to normalize NUP153-DamID-Seq data. We identified 73,018 high confidence NUP153 binding

sites (greater than 2-fold enrichment over Dam-only control and FDR < 0.05) (Supplementary Data 1). In agreement with an earlier report[26], we detected 32.2% of the NUP153 peaks at intergenic sites, 14.2% of peaks at promoters, and 53.5% of peaks across gene bodies (Fig. 2a).

We next examined NUP153 distribution across various genetic elements (Fig. 2b). We found that NUP153 is enriched at the TSS (Supplementary Fig. 2a) and 31.5% of TSS are NUP153-positive (7721/24,513) (Fig. 2b and Supplementary Data 2a). To define the transcriptional state of NUP153-positive TSS, we performed RNA-Seq and utilized previously published Histone 3 Lysine 4 trimethylation (H3K4me3) and H3K27me3 ChIP-Seq data[40] (Supplementary Fig. 2b). We found that NUP153 occupied both transcriptionally active and inactive TSS with a bias towards the active genes (Supplementary Fig. 2b). To evaluate NUP153 binding across enhancers, we mapped enhancers (n = 16,242) using previously published ChIP-Seq against enhancer-specific histone marks, H3K4me1 (ref. [41]), Histone 3 Lysine 27 acetylation (H3K27Ac)[42], and Chromatin Binding Protein (CBP)/P300 (ref. [43]) (Supplementary Fig. 2c). We detected NUP153 enrichment at the enhancers (Supplementary Fig. 2c) and identified 17.5% NUP153-positive enhancers (2849/16,242) (Fig. 2b). Compared to NUP153-negative enhancers, NUP153-positive enhancers exhibited higher H3K4me1, H3K27Ac, and CBP/P300 occupancy (Supplementary Fig. 2d). TSS- and

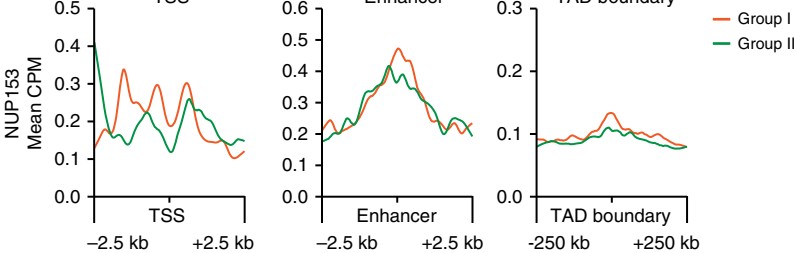

**a** Distribution of NUP153 peaks across the ES cell genome

- □ Promoter
- □ Intergenic
- ■ Gene body

**b**

— NUP153-positive
— NUP153-negative

TSS (*n* = 24,513)

Enhancer (*n* = 16,242)

TAD boundary (*n* = 5957)

| No. of sites | TSS | Enhancer | TAD boundary |
|---|---|---|---|
| NUP153-positive | 7721 (31.5%) | 2849 (17.5%) | 3984 (66.9%) |
| NUP153-negative | 16,792 (68.5%) | 13,393 (82.5%) | 1973 (33.1%) |

**c** CTCF sites Control

7034
2180
1679
830
1188
133
420

NUP153 KD-2
NUP153 KD-1

SMC3 sites Control

1762
154
76
61
205
5
65

NUP153 KD-2
NUP153 KD-1

**d** CTCF and SMC3 distribution across CTCF-positive sites

TSS (*n* = 2164)

Enhancer (*n* = 2272)

TAD boundary (*n* = 2238)

— Control
— NUP153 KD-1
— NUP153 KD-2

**e**

| | CTCF-positive sites | | | No. of NUP153 target genes |
|---|---|---|---|---|
| Group | TSS | Enhancer | TAD boundary | |
| I | 1123 | 1222 | 1144 | 558 |
| II | 1040 | 1049 | 1093 | 281 |

NUP153 distribution across Group I and II CTCF-positive sites in NUP153-Dam ES cells

TSS

Enhancer

TAD boundary

— Group I
— Group II

enhancer-specific NUP153 binding suggested a functional role for NUP153 in gene regulation. Given that NUP153 interacts with CTCF and cohesin, we also examined NUP153 binding across TAD boundaries[44]. We found NUP153 association with 66.9% of

TAD boundaries (3984/5957) (Fig. 2b and Supplementary Data 2) suggesting that NUP153 may functionally cooperate with CTCF and/or cohesin at the TAD boundaries during chromatin organization.

**Fig. 2 NUP153 mediates CTCF and cohesin binding at *cis*-regulatory elements and TAD boundaries in mouse ES cells. a** Distribution of NUP153 peaks in mouse ES cells. Peaks are categorized as promoters (−2 kb from TSS to +100 bp from TSS), gene body (+100 bp from TSS to +1 kb from transcription termination site (TTS)), intergenic sites (<−2 kb from TSS and >+1 kb from TTS). See Supplementary Data 1 for a list of NUP153 binding sites. **b** Metagene profiles of mean NUP153 binding at NUP153-positive and NUP153-negative TSS and enhancer (±5 kb), and TAD boundaries (±250 kb) (top). Number and percentage of NUP153 binding sites are presented as a table for the indicated genetic elements (bottom). (See Supplementary Data 2 for NUP153 binding sites at different genetic elements.) **c** Genome-wide CTCF and SMC3 binding sites were compared in control and NUP153 deficient (KD-1, KD-2) mouse ES cells. **d** Metagene profiles showing mean CTCF and SMC3 binding across CTCF-positive TSS ($n = 2164$), enhancers ($n = 2272$), and TAD boundaries ($n = 2238$) in control and NUP153 deficient mouse ES cells. **e** Mean CTCF binding in control and NUP153 KD mouse ES cells were compared and CTCF sites were grouped into two. Group I contained CTCF sites that showed greater mean CTCF binding in control cells over NUP153 KD cells. Group II contained CTCF sites that showed equal or lesser mean CTCF binding in control cells over NUP153 KD cells. Number of CTCF-positive sites across TSS and enhancer (±2.5 kb) and TAD boundaries (±250 kb), and the number of NUP153 target genes that associate with each group are shown as a table (top) (see also Supplementary Data 4 and Supplementary Fig. 3d). Metagene profiles showing mean NUP153 binding across CTCF-positive Group I and Group II TSS, enhancer (±2.5 kb) and TAD boundaries (±250 kb) (bottom).

## NUP153 mediates CTCF and cohesin binding at *cis*-regulatory elements and TAD boundaries.

To determine the functional relevance of NUP153 interaction with CTCF and cohesin, we mapped CTCF and cohesin binding sites by ChIP-Seq. In accordance with earlier reports[45], we detected CTCF and SMC3 across TSS, enhancers (Supplementary Fig. 2a, c) and TAD boundaries (Fig. 2d). We found that on average CTCF and cohesin binding sites were at ~5 kb distance with respect to the nearest NUP153 binding sites (Supplementary Fig. 2e). Based on this criterion, we detected a robust co-localization, whereby 48.9% of the CTCF and 44.4% of the SMC3 binding sites were co-occupied by NUP153 (Supplementary Data 3). Out of the CTCF and NUP153 co-occupied sites (CTCF+/NUP153+), 29.9% associated with TSS, and 24.2% associated with enhancers. Cohesin and NUP153 co-occupied sites (SMC3+/NUP153+) presented a similar profile in that 23.9% of these sites associated with TSS, and 27.1% associated with enhancers. We found that 10.4% of the TSS and 13.9% of the enhancers showed binding for all three factors (Supplementary Data 3). These results pointed to a potential crosstalk between NUP153, CTCF, and cohesin during the regulation of gene expression and/or chromatin architecture.

We next investigated the regulatory role of NUP153 in CTCF and cohesin binding by performing ChIP-Seq in control and NUP153-deficient mouse ES cells. To generate NUP153-deficient ES cells, we transduced cells with two different mouse NUP153-specific shRNA lentivirus (Supplementary Fig. 2f). NUP153 knockdown (KD) cells showed typical pluripotent ES cell characteristics with normal morphology and the presence of alkaline phosphatase activity, suggesting that NUP153 depletion did not interfere with the pluripotent state of ES cells (Supplementary Fig. 2g). By utilizing an oligo (dT)50-mer probe and performing RNA fluorescent in situ hybridization (FISH)[46], we further validated that the Poly(A)+ RNA export function of the NPCs was intact in NUP153 KD ES cells (Supplementary Fig. 2g).

CTCF and cohesin ChIP-Seq revealed that NUP153 depletion leads to a significant loss of CTCF (~60%) and SMC3 (~86%) binding genome-wide (Fig. 2c). NUP153 KD cells exhibited higher CTCF binding at promoters (30–34%) and lower CTCF binding across the gene bodies (38–41%), and intergenic sites (27–29%) (Supplementary Fig. 3a). Similar distribution patterns were detected for cohesin in NUP153 KD ES cells (Supplementary Fig. 3a). These results suggested that NUP153 may impact CTCF and cohesin binding across the genome.

Given that cohesin binding has been suggested to rely on CTCF[47], we focused on CTCF binding sites and showed that NUP153 is enriched at the CTCF-positive-TSS ($n = 2164$; $p = 0$, hypergeometric test), -enhancers ($n = 2272$; $p = 0$, hypergeometric test) and -TAD boundaries ($n = 2238$; $p = 8.66e{-}103$, hypergeometric test) (Supplementary Fig. 3b). Notably, NUP153 depletion resulted in reduction in CTCF and cohesin binding across the CTCF-positive genetic elements (Fig. 2d). To

determine how NUP153 binding influences CTCF distribution, we calculated the mean CTCF binding in control and NUP153 KD cells and grouped the CTCF binding sites into two. Group I contained CTCF sites that showed greater mean CTCF binding in control cells over NUP153 KD cells. Group II contained CTCF sites that showed equal or lesser mean CTCF binding in control cells over NUP153 KD cells (Fig. 2e, Supplementary Fig. 3c–e, and Supplementary Data 4). Group I TSS sites constituted ~10% (1123/11,726) of the total CTCF binding sites and half of these sites (~5%, 558/11,726) were NUP153 positive. Notably, metagene profiles across TSS, enhancer and TAD boundaries at Group I sites showed higher NUP153 binding compared to Group II sites (Fig. 2e and Supplementary Data 5). This data suggested that the degree of NUP153 binding correlates with differential change in CTCF binding at each genetic element. We concluded that NUP153 mediates CTCF and cohesin binding at TSS, enhancer, and TAD-boundaries. These findings raised the possibility that NUP153 may be critical for enhancer–promoter functions or chromatin organization functions of CTCF and cohesin during gene expression.

## NUP153 mediates transcription at bivalent genes and genome-wide.

To determine the extent of transcriptional changes in NUP153 deficient mouse ES cells, we performed RNA-Seq. NUP153 depletion resulted in differential expression of 711 genes (fold change ≥ 1.5 and FDR < 0.05) genome-wide (Fig. 3a and Supplementary Data 6). Approximately 56% (398/711) of the differentially regulated genes displayed NUP153 binding at TSS or gene bodies. A majority of the differentially regulated genes (66.2%, 471/711) were upregulated in NUP153 KD ES cells. Gene ontology (GO) analysis has revealed that the upregulated genes were associated with pathways such as those that impact cell differentiation (e.g., *Fgf1*, *Fgf9*, *Dlk1*, *Bmp7*, *Hoxb13*), and transcription regulation (e.g., *Wnt7b*, *Gata3*, *Bcl11a*, *ApoB*, *Lhx1*, *Pou3f2*) (Supplementary Data 7). By contrast, expression of genes that regulate biological processes such as extracellular matrix organization (e.g., *Fbln5*, *Comp*, *Ntn4*, *Dmp1*), response to mechanical stimulus (e.g., *Cav1*, *Cxcl12*, *Col3a1*), and skeletal muscle development (e.g., *Mef2c*, *Foxp2*, *Meox2*) were downregulated in NUP153-deficient ES cells.

We next investigated how NUP153-dependent changes in CTCF binding impact transcription. We found that ~34.4% (245/711) of the differentially regulated genes associated with CTCF-positive TSS (Supplementary Data 8). Majority of these genes (~61%) showed transcriptional upregulation in NUP153 KD mouse ES cells (Fig. 3b and Supplementary Data 8). GO analysis has revealed that these genes associate with important cellular processes such as the cell migration (e.g., *Ptk2b*, *Tcaf2*, *Wnt11*), cell adhesion (e.g., *Alcam*, *App*, *Itga3*, *Itga8*, *PLCb1*), and cell differentiation (e.g., *Foxa3*, *Flnb*, *Zfp423*, *Tnk2*) (Supplementary Data 7). Because CTCF-positive Group I sites

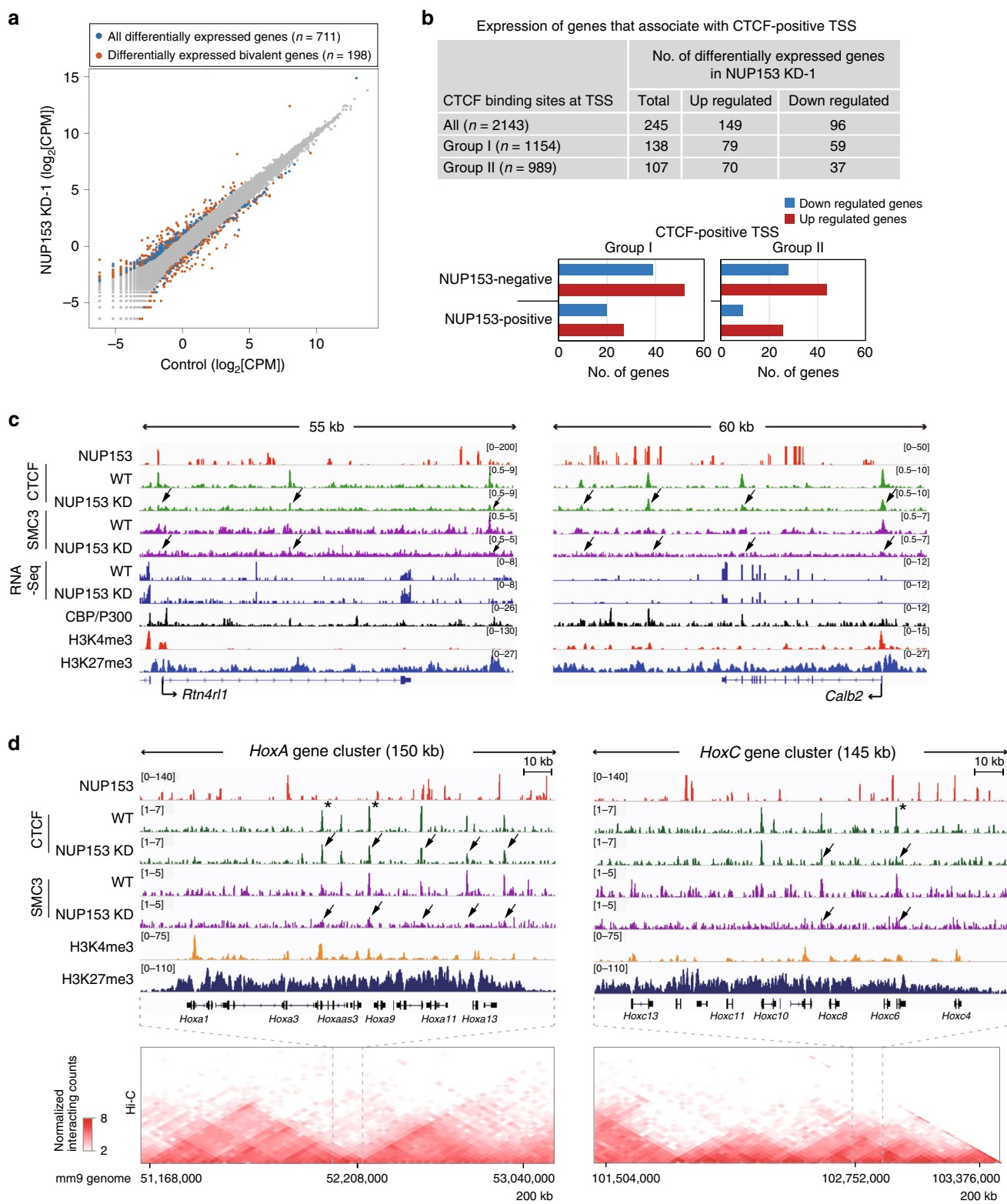

**b** Expression of genes that associate with CTCF-positive TSS

| | No. of differentially expressed genes in NUP153 KD-1 | | |
|---|---|---|---|
| CTCF binding sites at TSS | Total | Up regulated | Down regulated |
| All (n = 2143) | 245 | 149 | 96 |
| Group I (n = 1154) | 138 | 79 | 59 |
| Group II (n = 989) | 107 | 70 | 37 |

were enriched for NUP153 compared to Group II sites, and showed drastic change in CTCF binding in NUP153 KD mouse ES cells, we evaluated the number of differentially regulated genes between two groups. Group I sites associated with 19.4% (138/711) and Group II sites associated with 15% (107/711) of the differentially regulated genes in NUP153 KD-1 mouse ES cells (Fig. 3b). Notably, NUP153-positive Group I genes constituted ~7% (47/711) of the differentially regulated

genes. Within this group, upregulated (57.4%, 27/47) and downregulated (42.6%, 20/47) genes were almost equally distributed. Representative tracks shown for NUP153-positive Group I genes, *Rtn4rl1* and *Calb2*, in Fig. 3c present differential expression and altered CTCF and cohesin binding in NUP153 KD mouse ES cells (Fig. 3c). Collectively, these data suggested a regulatory role for NUP153 in global gene expression and for ~7% (47/711) of the differentially expressed

**Fig. 3 NUP153 influences transcription and binding of CTCF and cohesin at bivalent genes. a** Scatter plot showing expression levels of transcripts in log2 [CPM] scale in control and NUP153 KD-1 mouse ES cells. Blue points denote all differentially expressed genes ($n = 711$) and orange points denote differentially expressed bivalent genes (Supplementary Data 6 and 9). **b** Table showing number of differentially expressed genes that associate with all, Group I and Group II CTCF-positive TSS (top). Plots showing number of differentially regulated NUP153-positive and NUP153-negative genes that associate with Group I and II CTCF-positive TSS (bottom) (see also Supplementary Data 8). **c** NUP153 DamID-Seq, CTCF, cohesin, H3K4me3, and H3K27me3 ChIP-Seq, and RNA-Seq tracks are shown for two NUP153-positive Group I genes, *Rtn4rl1* (left panel) and *Calb2* (right panel) in control (WT) and NUP153 KD ES cells. *Rtn4rl1* shows transcriptional upregulation and *Calb2* shown transcriptional downregulation. **d** NUP153 DamID-Seq, CTCF, cohesin, H3K4me3, and H3K27me3 ChIP-Seq tracks are shown for a 145–150 kb region for the *HoxA* and *HoxC* loci in control (WT) and NUP153 KD mouse ES cells as indicated. Arrows point to regions where CTCF or SMC3 binding are altered in NUP153 KD mouse ES cells. CTCF sites labeled with asterisk (*) denote CTCF sites that have been reported to regulate transcription at the *Hox* loci by mediating the formation of TADs[48,70]. The 2D heat map shows the interaction frequency in mouse ES cells[44]. Hi-C data was aligned to the mm9 genome showing *HoxA* cluster residing in a TAD boundary and *HoxC* cluster in a TAD as published[44]. H3K4me3 and H3K27me3 (ref. [40]) and CBP/P300 (ref. [43]) ChIP-Seq data were previously published. CPM, counts per million.

genes this function underlies NUP153-mediated CTCF binding at TSS.

Bivalent state of genes has been proposed to be critical for establishment and maintenance of the ES cell pluripotency transcription program[1]. Recent evidence suggests that chromatin organization impacts the maintenance of bivalent state[4]. We thus envisioned that NUP153 may influence bivalent gene transcription by mediating CTCF or cohesin binding. We utilized the bivalent gene list ($n = 3868$) reported by Mas et al.[4] and identified 27.8% (198/711) of differentially regulated bivalent genes in NUP153 KD mouse ES cells ($p = 1.21e−16$, hypergeometric test) (Fig. 3a and Supplementary Data 9). Of this gene set, 32.8% (65/198) contained NUP153 binding, and ~10% (20/198) associated with NUP153-positive Group I TSS sites (Supplementary Data 8). This data supported a key role for NUP153 in transcription of bivalent genes and suggests that expression of a small proportion of bivalent genes is mediated through NUP153-mediated CTCF binding at TSS.

This analysis has also revealed that NUP153 associates with several *Hox* genes, which are characterized as bivalent genes in mouse ES cells[1,48]. Genomic organization of the *Hox* loci relies on TADs with enriched CTCF binding[44] and influences developmental expression of *Hox* genes[49]. As presented in the representative tracks shown for the *HoxA* and *HoxC* clusters, we found that NUP153 depletion resulted in altered CTCF and/or cohesin binding at specific *Hox* genes (Fig. 3d, arrows). Importantly, three of these CTCF-binding sites (Fig. 3d, asterisks) have been reported to be critical in facilitating the formation of TADs and providing an insulator function during *Hox* gene transcription in mouse[48]. Based on these data, we propose that NUP153 may contribute to the higher-order chromatin organization by regulating CTCF and cohesin binding at specific developmental genes, such as the *Hox* loci, and mediates their gene expression.

**NUP153-mediated POL II recruitment is critical for timely IEG transcription.** To provide a mechanistic understanding on NUP153-mediated gene expression and the interplay between NUP153, and CTCF and cohesin, we utilized EGF-inducible IEGs[50]. Several characteristics of these loci suggested that they would provide a powerful in vivo model for our studies. First, we identified that IEGs, *Egr1*, *c-Fos*, and *Jun*, are NUP153 targets (Supplementary Data 1). Second, TSS and distal regulatory elements of IEGs showed CTCF and cohesin occupancy in mouse ES cells (Supplementary Fig. 4). Third, during the preparation of this manuscript, it was shown that CTCF-mediated higher-order chromatin structure impacts transcription of IEGs[51,52]. Lastly, due to their inducible nature, the IEG loci can be utilized for mechanistic studies to elucidate NUP153-mediated chromatin structure during transcriptional silencing and activation. To test the regulatory role for NUP153 in IEG transcription, we could

not use mouse ES cells because IEG transcription kinetics show variability in these cells and thus could not be stably measured[53]. We thus utilized HeLa cells in which IEG transcription can be reduced to a silent state by serum starvation and transcription initiation can be reproducibly induced by EGF treatment[29]. We generated NUP153 KD HeLa cells by shRNA lentivirus (Fig. 4a) and validated that NUP153 knockdown did not alter the nucleocytoplasmic trafficking at the NPCs by quantitating the nuclear import and export of the dexamethasone (Dex) responsive GFP-tagged glucocorticoid receptor (GR)[54] (Supplementary Fig. 5a, b). Furthermore, as in mouse ES cells, NUP153 KD HeLa cells did not present any defects in Poly(A)$^+$ RNA export (Supplementary Fig. 5c).

To evaluate NUP153-dependent changes in IEG transcription, we utilized *EGR1*, *JUN*, *c-FOS* genes, and assessed transcription induction in response to EGF treatment in control and NUP153 deficient HeLa cells in a time course dependent manner. We found that NUP153 depletion led to a significant reduction in IEG mRNA and pre-mRNA levels upon 15 min of EGF treatment compared to control cells (Fig. 4b and Supplementary Fig. 6a). This effect was NUP153-specific, as expression of FLAG-NUP153 in NUP153-deficient HeLa cells led to the recovery of transcription initiation (Fig. 4c). At 30 min EGF treatment, *EGR1* and *c-FOS* pre-mRNA levels were significantly upregulated in NUP153 deficient cells (Fig. 4b and Supplementary Fig. 6a). This data suggested that the suppression of IEG transcription during the initiation step may lead to a delay in transcription or trigger a passive negative feedback on IEG transcription[29]. These data collectively indicated that NUP153 acts as an activator of IEG transcription.

Based on our findings, we reasoned that NUP153 may control POL II occupancy during IEG transcription. To investigate, we performed POL II ChIP and quantitatively measured POL II occupancy at the TSS and across gene bodies (GB) of *JUN* and *EGR1* using gene-specific primers (Fig. 4d and Supplementary Data 10). We found that POL II binding at the TSS of IEGs was significantly reduced in NUP153 KD HeLa cells during the paused state (minus EGF). Furthermore, the expected POL II enrichment across the TSS and the gene bodies was significantly altered in NUP153 KD HeLa cells upon induction of transcription (15 min EGF). By contrast, POL II binding across the IEGs was comparable between NUP153 KD and control HeLa cells at 30 min of EGF induction. These results were in line with data showing that NUP153 is critical for timely IEG transcription initiation (Fig. 4b). We concluded that NUP153 regulates IEG transcription initiation by controlling POL II occupancy at the TSS during the paused state.

**NUP153 controls CTCF and cohesin binding at the IEG *cis*-regulatory elements.** To define how NUP153 influences IEG-specific changes in CTCF and cohesin binding, we examined

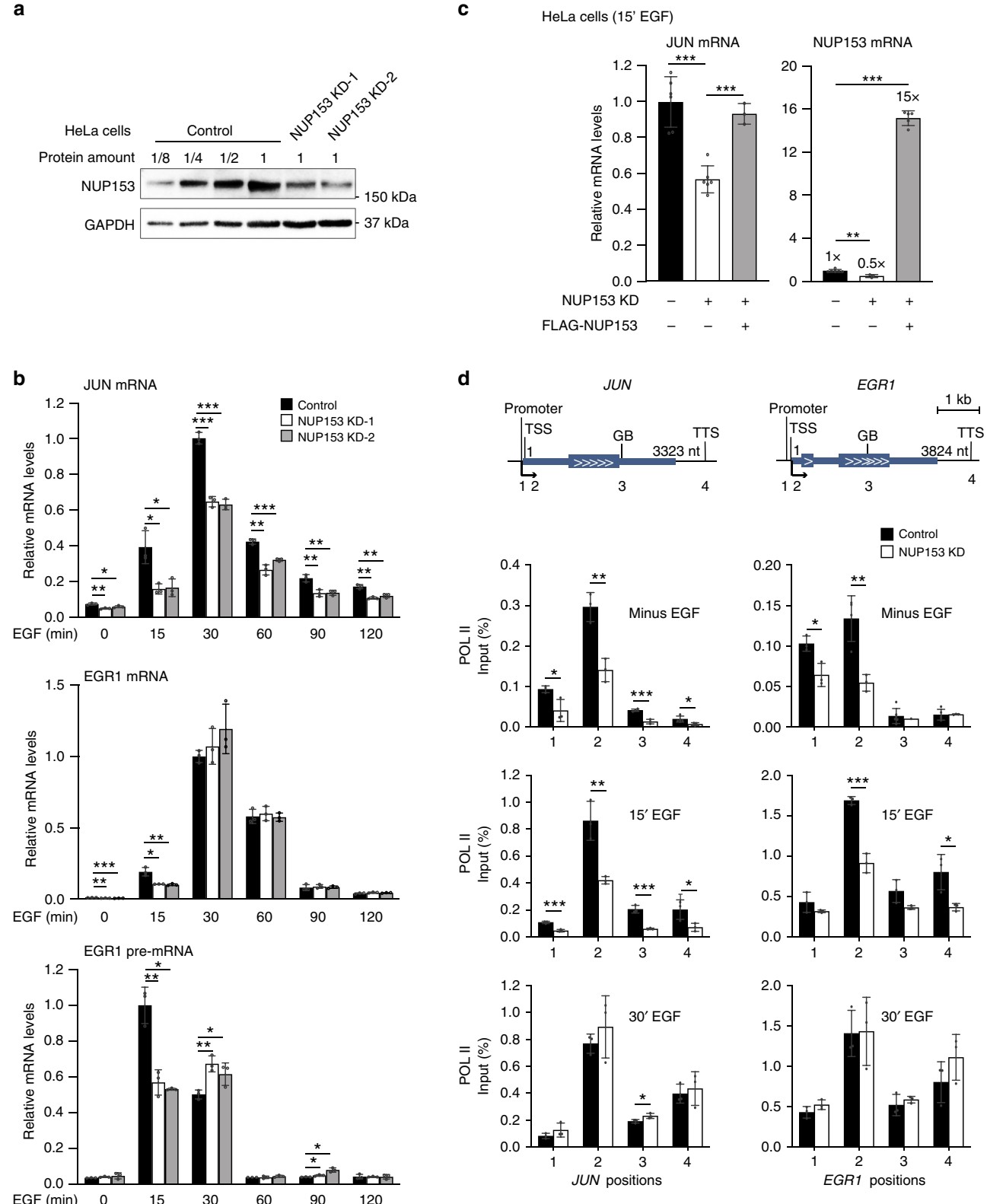

CTCF, cohesin, and POL II binding along with chromatin structure at the *EGR1* and *JUN* loci using ENCODE ChIP-Seq data in HeLa cells[55] (Supplementary Fig. 6b). This analysis allowed us to design IEG-specific primers (Supplementary Data 10) across distal enhancers, TSS, GB, and transcription termination sites (TTS), and quantitatively determine NUP153, cohesin, and CTCF binding in a time-course dependent manner.

NUP153 ChIP revealed that NUP153 associates with the *EGR1* and *JUN* distal enhancers, TSS and TTS at the paused state (minus EGF) (Fig. 5). Interestingly, we found that NUP153 associates across these loci in a transcription-dependent manner (15 min EGF, Fig. 5) suggesting a tight coupling between NUP153 binding and transcriptional state. Similar to NUP153, CTCF and cohesin binding was enriched around enhancers in control

**Fig. 4 NUP153 controls POL II recruitment to the IEG promoters and impacts IEG transcription initiation. a** Western blot showing NUP153 protein levels in control and NUP153 KD HeLa cells. **b** Real-time RT-PCR showing relative IEG mRNA and nascent mRNA levels in control and NUP153 KD HeLa cells in a time course dependent manner. JUN mRNA levels at minus EGF (control vs KD-1, **$p = 0.0033$; control vs KD-2, *$p = 0.0345$), 15 min EGF (control vs KD-1, *$p = 0.0138$; control vs KD-2, *$p = 0.0199$), 30 min EGF (control vs KD-1, ***$p = 0.0001$; control vs KD-2, ***$p = 0.0001$), 60 min EGF (control vs KD-1, **$p = 0.0010$; control vs KD-2, ***$p = 0.0004$), 90 min EGF (control vs KD-1, **$p = 0.0065$; control vs KD-2, **$p = 0.0045$), 120 min EGF (control vs KD-1, **$p = 0.0010$; control vs KD-2, **$p = 0.0050$). EGR1 mRNA levels at minus EGF (control vs KD-1, **$p = 0.0018$; control vs KD-2, ***$p = 0.0008$), 15 min EGF (control vs KD-1, *$p = 0.0373$; control vs KD-2, **$p = 0.0066$). EGR1 pre-mRNA levels at 15 min EGF (control vs KD-1, **$p = 0.0038$; control vs KD-2, *$p = 0.0151$), 30 min EGF (control vs KD-1, **$p = 0.0045$; control vs KD-2, *$p = 0.0436$), 90 min EGF (control vs KD-1, *$p = 0.0364$; control vs KD-2, *$p = 0.0305$). **c** Real-time RT-PCR showing relative JUN and NUP153 mRNA levels in control, NUP153 KD and FLAG-NUP153 expressing NUP153 KD HeLa cells upon 15 min EGF treatment. GAPDH was used to normalize mRNA levels. JUN mRNA levels in control vs KD (***$p = 0.0000$); KD vs FLAG-NUP153 (***$p = 0.0001$). NUP153 mRNA levels in control vs KD (**$p = 0.0011$); KD vs FLAG-NUP153 (***$p = 0.0000$). **d** POL II binding across *JUN* and *EGR1* genetic elements was mapped by ChIP real-time PCR in control and NUP153 KD HeLa cells under indicated conditions (see Supplementary Data 10 for primer sequences). POL II binding at *JUN*, minus EGF (control vs KD; promoter, *$p = 0.0329$; TSS, **$p = 0.0042$; GB, ***$p = 0.0003$; TTS, *$p = 0.0493$), 15 min EGF (control vs KD; promoter, ***$p = 0.0009$; TSS, **$p = 0.0065$; GB, ***$p = 0.0009$; TTS, *$p = 0.0443$), 30 min EGF (control vs KD; GB, *$p = 0.0385$). POL II binding at *EGR1*, minus EGF (control vs KD; promoter, *$p = 0.0172$; TSS, **$p = 0.0060$), 15 min EGF (control vs KD; TSS, ***$p = 0.0004$; TTS, *$p = 0.0281$). Data shown are percent (%) of input. Values are mean ± standard deviation. Two-tailed Student's *t*-test, $n \geq 3$ independent experiments. Nt, nucleotide. Source data are provided as a Source Data file.

HeLa cells at the paused state and upon transcriptional activation with EGF, both proteins dynamically dissociated from these sites. In NUP153-deficient HeLa cells, CTCF and cohesin binding at the enhancers were significantly reduced at the paused state, but did not change upon transcriptional activation (15 min EGF) (Fig. 5). These results suggested that enhancer-specific binding of both proteins relies on NUP153. Given that *EGR1* transcription relies on CTCF-mediated higher-order chromatin[52], it is likely that NUP153 influences IEG chromatin organization by mediating CTCF and cohesin binding during the paused state.

**Co-regulatory function of NUP153 and CTCF during the IEG paused state.** Based on these results, we hypothesized that NUP153-mediated CTCF and cohesin binding at the IEG enhancers might be necessary for the proximal-promoter binding of POL II during the IEG paused state. We specifically focused on the functional relationship between NUP153 and CTCF because cohesin distribution depends on CTCF binding and POL II elongation[56]. We generated CTCF knockdown HeLa cells by using shRNA (Fig. 6a). Similar to what we detected in NUP153 KD HeLa cells (Fig. 4b), CTCF depletion resulted in significant decrease in the TSS- and promoter-specific POL II binding during the IEG paused state and reduction in the IEG transcription initiation (Fig. 6b, c). Importantly, targeting both NUP153 and CTCF by shRNA (NUP153/CTCF KD) in HeLa cells did not cause an additive effect in downregulation of IEG transcription (Fig. 6d). This data suggests that NUP153 and CTCF mediate IEG transcription through the same regulatory mechanism.

**NUP153-dependent spatial positioning of IEGs during transcription regulation.** Here, we investigated the spatial positioning of the *c-FOS* locus and its dependency on NUP153 in a time course dependent manner. To this end, we performed *c-FOS* DNA FISH in combination with LAMIN B1 immunofluorescence and examined the sub-nuclear position of *c-FOS* DNA with respect to the nuclear periphery in control and NUP153 KD HeLa cells (Fig. 7a). Analysis of cumulative frequency graphs has revealed that *c-FOS* locus is closely positioned (ND ≤ 0.12) to the nuclear periphery in ~30% of the control cells at the paused state (minus EGF) and that the loci moved even closer to the periphery (ND ≤ 0.10) upon transcription induction (Fig. 7b). By contrast, the locus remained distal to the periphery independent of the transcriptional state in NUP153 KD HeLa cells (Fig. 7b and Supplementary Fig. 7). These results argue that NUP153-dependent positioning of IEG to the NPC is critical during

transcription regulation and suggest that NUP153 mediates spatial positioning of CTCF and cohesin to the NPC during IEG transcription.

**Discussion**

In this study, we aimed to provide a mechanistic understanding on how NUP153 mediates chromatin structure and influences transcription. We identified NUP153 association with the chromatin architectural proteins, CTCF and cohesin, and revealed that NUP153 is a critical regulator of chromatin structure and transcription by affecting CTCF and cohesin binding across *cis*-regulatory elements and TAD boundaries. Even though we cannot exclude the fact that nucleoplasmic or nuclear matrix association is not possible, our findings suggest that the co-regulatory function of NUP153 and architectural proteins likely occurs around the NPC (Figs. 1d, 7 and Supplementary Fig. 7). Our findings are in line with earlier reports in yeast and Drosophila showing that the inducible genes associate with the NPC and are mediated through chromatin looping between distal regulatory elements and promoters[13–15].

NUP153 has been associated with cell-type-specific transcription[27] and implicated in chromatin accessibility[28]. Similarly, CTCF exhibits variable binding patterns influencing cell-type-specific transcription[57]. It is thus plausible to speculate that NUP153 might cooperate with CTCF in higher-order chromatin organization in the regulation of cell-type-specific transcription. We propose that bivalent genes might be under such control. This is because bivalent genes are mediated through the simultaneous catalytic activity of MLL and Polycomb Repressive Complex 2 (PRC2). Recent work suggests that regulation of chromatin organization is equally important[4]. Specifically, MLL2 deficiency in mouse ES cells results in increased Polycomb binding and loss of chromatin accessibility at promoters, coupled with alterations in long-range chromatin interactions[4]. Investigating the co-regulatory function of NUP153 and CTCF in bivalency and cell-type-specific gene expression using ES cells are thus interests for future studies.

CTCF and cohesin have been shown to mediate insulation of TADs[16,17]. Nevertheless, depletion of either protein does not result in disappearance of all TADs pointing to a hierarchical control of higher-order chromatin organization[16,17,58]. In addition to TADs, chromatin compartments can be also established based on specific chromatin interactions with the lamina or the NPC[23]. NUP153 is among the NPC components that participate in such regulation. For example, the yeast NUP153 homologue

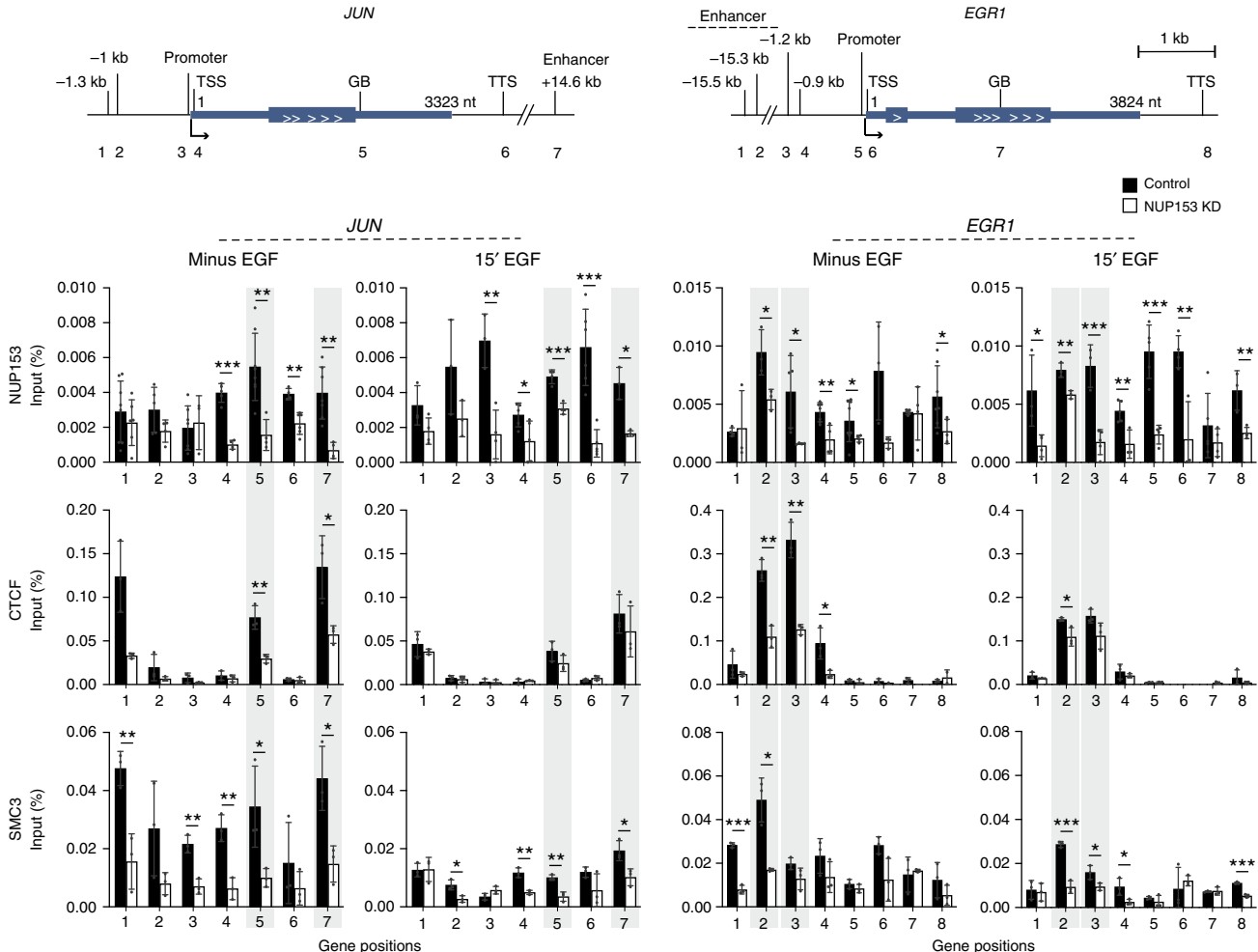

**Fig. 5 NUP153 is critical for CTCF and cohesin binding at the IEG *cis*-regulatory sites and across the IEG loci during the paused state.** NUP153, CTCF, and SMC3 occupancy across the *JUN* (left) and *EGR1* (right) genetic elements were examined by ChIP real-time PCR at the indicated time points in control and NUP153 KD HeLa cells. Position of PCR primers are denoted as numbers in the schematics (see Supplementary Data 10 for primer sequences). NUP153 binding at *JUN*, minus EGF (control vs KD; TSS, \*\*\**p* = 0.0000; GB, \*\**p* = 0.0035; TTS, \*\**p* = 0.0011; +14.6 kb, \*\**p* = 0.0023), 15 min EGF (control vs KD; promoter, \*\**p* = 0.0047; TSS, \**p* = 0.0368; GB, \*\*\**p* = 0.0003; TTS, \*\*\**p* = 0.0004; +14.6 kb, \**p* = 0.0301). NUP153 binding at *EGR1*, minus EGF (control vs KD; −15.3 kb, \**p* = 0.0298; −1.2 kb, \**p* = 0.0323; −0.9 kb, \*\**p* = 0.0039; promoter, \**p* = 0.0469; TTS, \**p* = 0.0406), 15 min EGF (control vs KD; −15.5 kb, \**p* = 0.0299; −15.3 kb, \*\**p* = 0.0068; −1.2 kb, \*\*\**p* = 0.0006; −0.9 kb, \*\**p* = 0.0086; promoter, \*\*\**p* = 0.0006; TSS, \*\**p* = 0.0078; TTS, \*\**p* = 0.0085). CTCF binding at *JUN*, minus EGF (control vs KD; GB, \*\**p* = 0.0047; +14.6 kb, \**p* = 0.0231). CTCF binding at *EGR1*, minus EGF (control vs KD; −15.3 kb, \*\**p* = 0.0017; −1.2 kb, \*\**p* = 0.0010; −0.9 kb, \**p* = 0.0291), 15 min EGF (control vs KD; −15.3 kb, \**p* = 0.0316). SMC3 binding at *JUN*, minus EGF (control vs KD; −1.3 kb, \*\**p* = 0.0076; promoter, \*\**p* = 0.0032; TSS, \*\**p* = 0.0037; GB, \**p* = 0.0413; +14.6 kb, \**p* = 0.0158), 15 min EGF (control vs KD; −1 kb, \**p* = 0.0154; TSS, \*\**p* = 0.0029; GB, \*\**p* = 0.0036; +14.6 kb, \**p* = 0.0245). SMC3 binding at *EGR1*, minus EGF (control vs KD; −15.5 kb, \*\*\**p* = 0.0001; −15.3 kb, \**p* = 0.0309), 15 min EGF (control vs KD; −15.3 kb, \*\*\**p* = 0.0004; −1.2 kb, \**p* = 0.0355; −0.9 kb, \**p* = 0.0419; TTS, \*\*\**p* = 0.0003). Data shown are percent (%) of input. Values are mean ± standard deviation. Two-tailed Student's *t*-test, *n* ≥ 3 independent experiments. Nt, nucleotide. Source data are provided as a Source Data file.

Nup2 acts as an insulator at the nuclear basket[59] and the mammalian NUP153 impacts establishment of heterochromatin domains in interphase cells[60]. Furthermore, NUP153 have been recently implicated in the compartmentalization of transcription factors at the NPC in response to the activation of signal transduction pathways during cellular senescence, cell migration, and cell proliferation[60,61]. Our results suggest that NUP153 may have a role in multistep organization and/or insulation of site-specific higher-order chromatin around the NPCs providing spatial and/or temporal organization of transcription in response to cellular cues (e.g., EGF signaling). The *Hox* loci and IEGs may be subjected to such regulation. We have determined that transcription of human IEGs and a subset (~5%) of the mouse ES cell genes rely on NUP153-mediated CTCF and/or cohesin binding at TSS.

Our results are in accordance with earlier findings showing that ~10% of all TSS bound CTCF associated with promoter activity[16]. Thus, future studies focusing on the role of NUP153 in chromatin structure and chromatin organization are critical.

Several genome-wide studies in metazoa have shown that the distribution of paused POL II shows a positive correlation with CTCF and cohesin binding[62]. CTCF is thought to induce POL II pausing by creating "roadblocks" on the DNA template obstructing transcription elongation[63]. Here, we provide new evidence that NUP153 cooperates with CTCF in the regulation of POL II occupancy at the IEGs during paused state and that NUP153 and CTCF mediate IEG transcription through the same regulatory mechanism. We propose that NUP153 interacts with CTCF and mediates its binding at the *cis*-regulatory elements

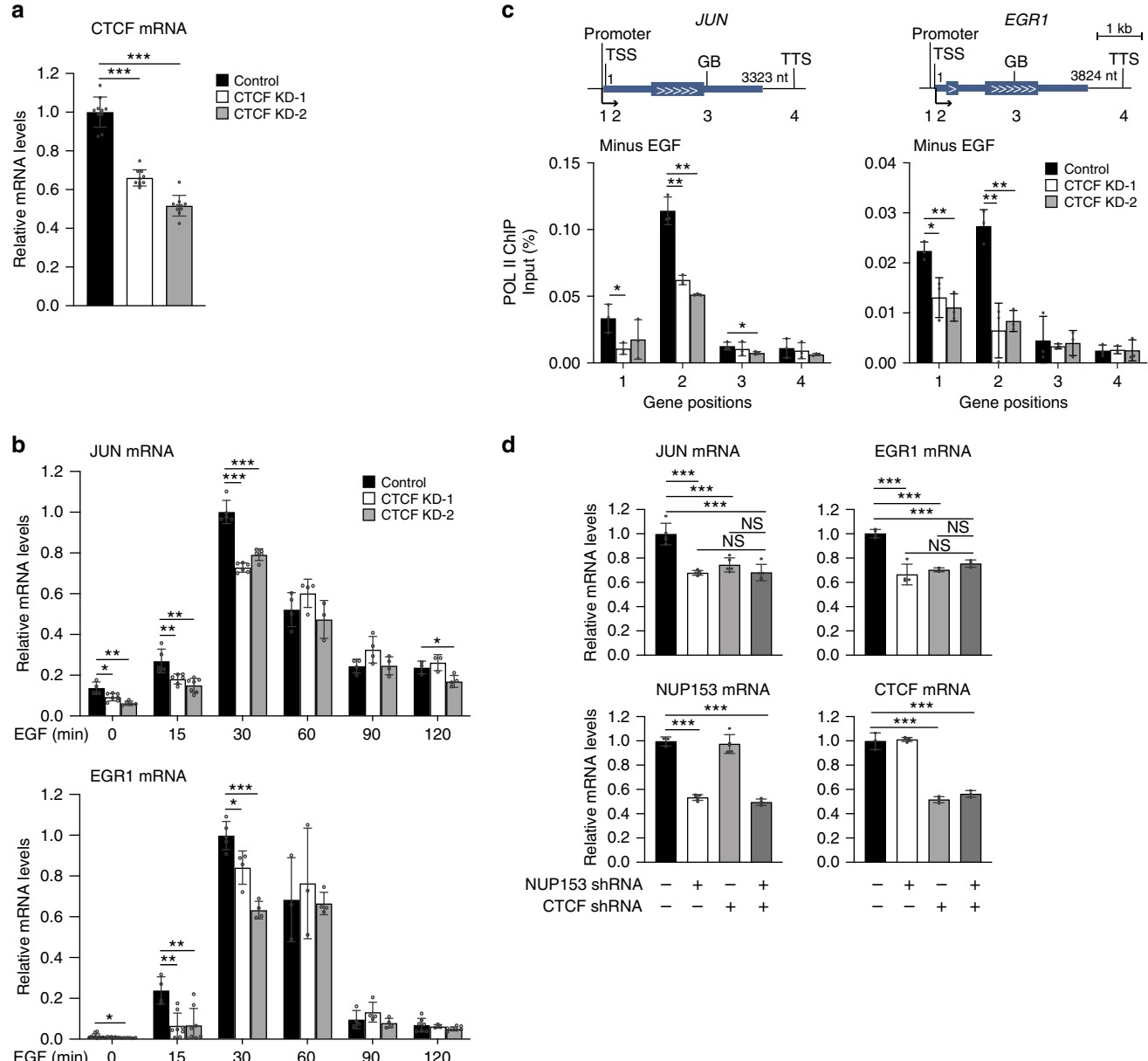

**Fig. 6 NUP153 and CTCF co-regulate POL II recruitment at the IEG paused state and impact IEG transcription. a** Real-time RT-PCR showing relative CTCF mRNA levels in HeLa cells transfected with control or CTCF shRNA expression vectors (KD-1 and KD-2). Control vs KD-1, ***$p = 0.0000$; control vs KD-2, ***$p = 0.0000$. **b** Real-time RT-PCR showing relative *JUN* and *EGR1* mRNA levels in control or CTCF KD HeLa cells. JUN mRNA levels at minus EGF (control vs KD-1, *$p = 0.0126$; control vs KD-2, **$p = 0.0011$), 15 min EGF (control vs KD-1, **$p = 0.0077$; control vs KD-2, **$p = 0.0011$), 30 min EGF (control vs KD-1, ***$p = 0.0000$; control vs KD-2, ***$p = 0.0000$), 120 min EGF (control vs KD-2, *$p = 0.0299$). EGR1 mRNA levels at minus EGF (control vs KD-2, *$p = 0.0473$), 15 min EGF (control vs KD-1, **$p = 0.0012$; control vs KD-2, **$p = 0.0050$), 30 min EGF (control vs KD-1, *$p = 0.0162$; control vs KD-2, ***$p = 0.0000$). **c** POL II binding across the IEGs, *JUN* and *EGR1* loci, was mapped by POL II ChIP in control and CTCF KD HeLa cells at the paused state (minus EGF). POL II occupancy at the indicated genetic elements was measured using primers as denoted in the schematics (Supplementary Data 10). POL II binding at *JUN* promoter (control vs KD-1; *$p = 0.0261$), TSS ((control vs KD-1; **$p = 0.0012$), (control vs KD-2; **$p = 0.0087$)), GB (control vs KD-2; *$p = 0.0463$); *EGR1* promoter ((control vs KD-1; *$p = 0.0209$), (control vs KD-2; **$p = 0.0038$)), TSS ((control vs KD-1; **$p = 0.0048$), (control vs KD-2; **$p = 0.0011$)). Data shown are percent (%) of input. **d** Real-time RT-PCR showing relative EGR1, JUN, NUP153, and CTCF mRNA levels in NUP153 KD, CTCF KD, and in CTCF/NUP153 KD HeLa cells. JUN mRNA levels (control vs NUP153 KD, ***$p = 0.0009$; control vs CTCF-KD, ***$p = 0.0006$; control vs NUP153 and CTCF KD, ***$p = 0.0002$). EGR1 mRNA levels (control vs NUP153 KD, ***$p = 0.0003$; control vs CTCF-KD, ***$p = 0.0000$; control vs NUP153 and CTCF KD, ***$p = 0.0001$). NUP153 mRNA levels (control vs NUP153 KD, ***$p = 0.0000$; control vs NUP153 and CTCF KD, ***$p = 0.0000$). CTCF mRNA levels (control vs CTCF-KD, ***$p = 0.0000$; control vs NUP153 and CTCF KD, ***$p = 0.0005$). Values are mean ± standard deviation. Relative mRNA levels were normalized using GAPDH. Two-tailed Student's *t*-test, $n \geq 3$ independent experiments. NS, not significant ($p > 0.05$). Nt, nucleotide. Source data are provided as a Source Data file.

which subsequently leads to cohesin recruitment and chromatin looping between gene regulatory elements and/or TADs at the NPC. This state is essential for the establishment of a poised chromatin environment at which efficient transcription initiation can be rapidly induced through a POL II pause–release mechanism in response to stimuli (Fig. 7c). Two recent reports showed that CTCF-mediated chromatin organization impacts IEG transcription[51,52] supporting our findings and proposed

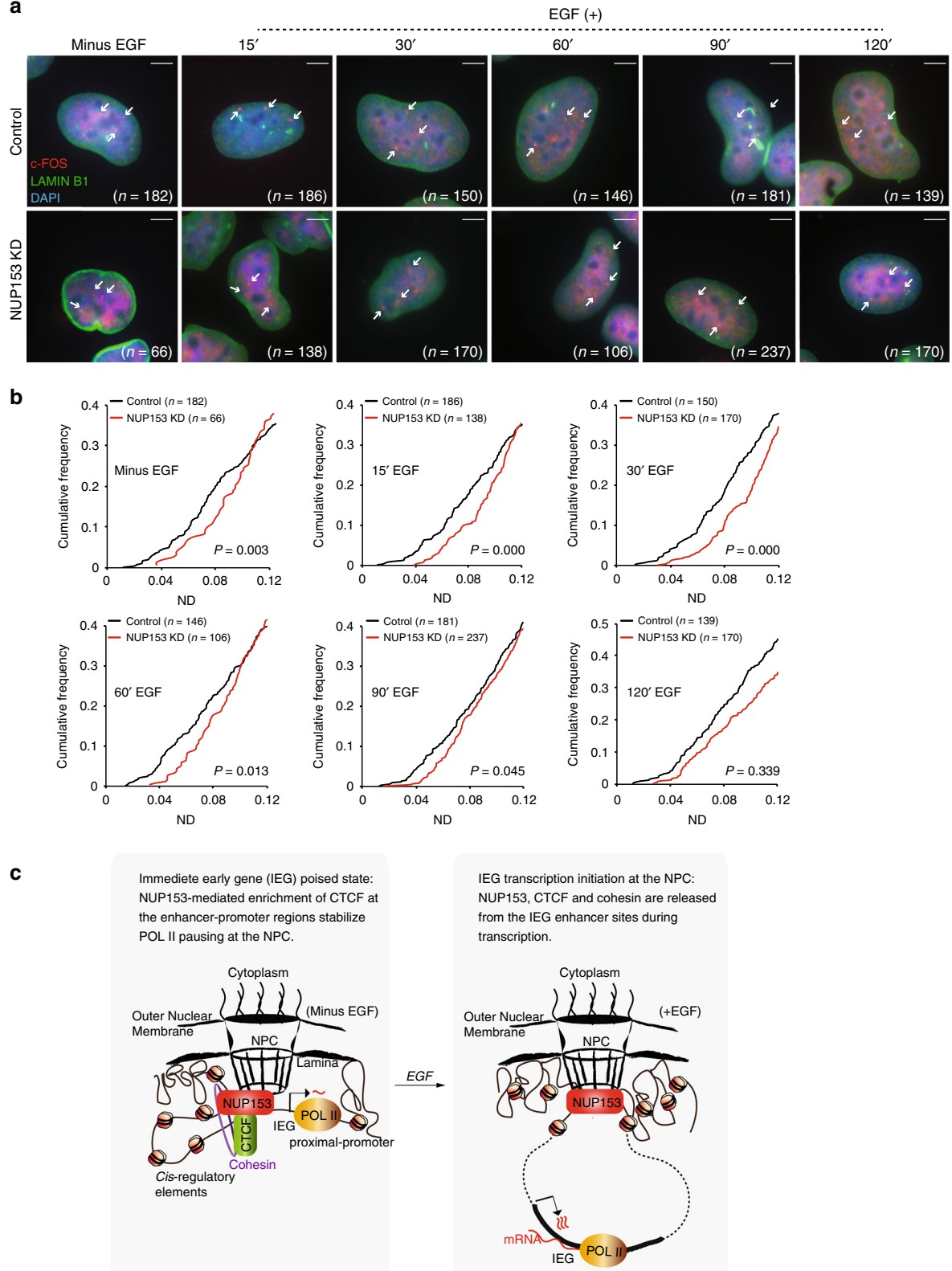

**Fig. 7 IEGs associate with the NPC in NUP153-dependent manner during paused state and transcription initiation. a** Immunostaining of LAMIN B1 and *c-FOS* DNA FISH in control and NUP153 KD HeLa cells are shown at the indicated time points. Cell numbers are as indicated. HeLa cells contain three *c-FOS* alleles (white arrows). Scale bar, 5 μm. **b** Cumulative frequency graphs showing distribution of the *c-FOS* locus distance to nuclear periphery in control and NUP153 KD HeLa cells at the indicated time points. Cumulative frequencies at a normalized distance (ND) of 0.0–0.12 are shown. See Supplementary Fig. 7 for the distribution of loci in all cells that were analyzed (ND of 0.0–0.5). ND = *c-FOS* locus to periphery distance/cell diameter (*d*), where *d* = (2× nuclear area/π)$^{0.5}$. *$p < 0.05$; ***$p < 0.001$; Kolmogorov–Smirnov (KS)-test, *n* = 2 independent experiments. **c** Working model showing NUP153-mediated chromatin structure and transcription regulation at the IEG locus. Source data are provided as a Source Data file.

model. NUP153-dependent localization to the NPC might thus provide an advantageous spatial position to genes that are poised to respond rapidly to developmental cues during ES cell pluripotency and/or differentiation. Furthermore, by examining NUP153 binding dynamics during transcription, we showed that NUP153 spreads across the IEG promoter and the gene bodies during transcriptional activation (Fig. 5). This data suggests that there might be a tight functional correlation between NUP153 and POL II activity during transcription. Chromatin sites that are engaged with stalled or active POL II might therefore allow for the differential NUP153 binding and can provide its selectivity towards transcriptionally silent or active chromatin domains.

We found that CTCF and cohesin binding sites were on average ~5 kb distance from the nearest NUP153 binding sites (Supplementary Fig. 2e). NUP153 may influence CTCF and cohesin binding directly or indirectly. One possible mechanism is through the scaffold feature of the NPCs[6]. Second possible mechanism might be through the establishment of an optimal chromatin environment at the putative CTCF binding sites by NUP153. CTCF-binding sites display characteristic chromatin structure showing DNase I hypersensitivity and enrichment of H3K4me3, H3K4me2, H3K4me1, and H2A.Z[64,65]. Thus, defining NUP153-interacting proteins and dissecting their co-regulatory function with NUP153 in chromatin structure can provide valuable insights on the underlying mechanisms of NUP153-mediated CTCF binding.

Our findings are also relevant towards the understanding of cancers that underlie defects in chromatin-associated function of Nups. Several Nups, including NUP153 and NUP98, contain unstructured phenylalanine–glycine (FG)-repeats[5]. Structural chromosomal rearrangements or translocations of the FG-Nup genes result in the formation of FG-Nup fusion proteins (e.g., NUP98-HOXD13, NUP98-HOXA9, NUP98-MLL), which have been implicated in several hematologic malignancies[66]. A recent report in Drosophila suggests that NUP98 forms a complex with several architectural proteins including CTCF[15]. Thus, we propose that enhancer-specific regulation of chromatin structure and organization by mammalian NUP153 may apply to other FG-Nups and contribute to the gene regulatory mechanisms that underlie FG-Nup fusion protein-associated cancers.

## Methods

**Cell culture, plasmids, virus preparation, and viral transduction.** EL16.7 female mouse ES cell line (gift from J.T. Lee (Harvard)) and cell culture conditions have been described previously[38]. Mouse ES cells were cultured on γ-irradiated mouse embryonic fibroblasts (MEFs) that were isolated from Tg(DR4)1Jae/J mice (The Jackson Laboratory). To transduce ES cells, control (scramble) or mouse NUP153 specific shRNA lentivirus particles (~$10^7$–$10^8$ TU/ml) were added into 0.5 ml of complete ES cell medium containing dissociated ES cells ($5 \times 10^5$), LIF (500 units/ml; ESGRO, Sigma-Aldrich) and Polybrene (4 μg/ml; Sigma-Aldrich) and incubated overnight at 37 °C. Next day, ES cells were dissociated and plated onto a 60-mm tissue culture dish (BD) containing γ-irradiated DR4 MEFs ($1 \times 10^6$), cultured for 24 h in regular ES cell media followed by 2 days of selection using 2 μg/ml puromycin (Puro) (Sigma-Aldrich) and collected for subsequent analyses. HEK293T and HeLa cells were obtained from the American Tissue Collection Center (ATCC, Manassas, VA, USA) through the Duke University Cancer Center Facilities and were maintained in high glucose Dulbecco's modified Eagle's medium (DMEM) GlutaMAX supplemented with 10% fetal bovine serum (FBS; Sigma-Aldrich), 1% penicillin/streptomycin, 1 mM sodium pyruvate, 1% non-essential amino acids, and 3% HEPES. To generate FLAG-NUP153 overexpressing cells, HEK293T cells were transfected with FLAG-mNUP153 or FLAG-hNUP153 cDNA vectors using Xfect reagent (Clonetech) according to the manufacturer's instructions. FLAG-hNUP153 (human) or FLAG-mNUP153 (mouse) expression vectors were constructed by amplifying full-length human NUP153 or mouse NUP153 cDNA using human NUP153 cDNA (Origene, SC116943) or mouse NUP153 cDNA (ATCC, IMAGE clone ID: 6516328) clones, respectively. Amplified cDNA sequences were modified and cloned into BamHI and XhoI sites of pCMV-3FLAG-6 vector (Agilent, 240200). To produce shRNA lentivirus particles, HEK293T cells were transfected with pMD2.G (Addgene #12259) and psPAX2 (Addgene #12260) vectors along with each shRNA lentiviral vector. Viral supernatants were

concentrated X100 using Lenti-X Concentrator (Clontech) according to the manufacturer's instructions, aliquoted and stored at −80 °C. All reagents were from Thermo Fisher Scientific, unless noted otherwise. All cells were cultured at 37 °C with 5% $CO_2$. Mouse husbandry and experiments were conducted in accordance with an approved protocol (A238-17-10) for the ethical use of animals in research by the Duke University Institutional Animal Care and Use Committee (IACUC).

**Generation of NUP153 mouse ES cell clones and NUP153 DamID-Seq.** Mouse Nup153 cDNA (4.5 kb) (IMAGE clone ID: 6516328; ATCC) was modified and cloned in frame into KpnI and XhoI sites in pIND-(V5)-EcoDam plasmid (gift from B. Van Steensel). To generate EcoDam or Nup153.EcoDam overexpressing mouse ES cells, 10 μg of NUP153-(V5)-EcoDam-pIND or (V5)-EcoDam-pIND plasmid DNA were introduced into wild-type EL16.7 mouse ES cells[38] ($1 \times 10^7$) by electroporation (200 V, 1050 μF) and stable clones were selected for 12 days using complete DMEM media supplemented with G418 (200 μg/ml) (Invitrogen). Positive clones were screened for EcoDam sequence by genomic DNA PCR using the primers: V5F, 5′-GGT AAG CCT ATC CCT AAC CCT C-3′; EcoDam_400R, 5′-AAC TCA CCG CGC AGA TTG TAA CG-3′ and by immunofluorescence as previously described[67]. Mouse monoclonal α-V5 antibody was used in combination with rabbit polyclonal anti-IgG(H+L)-Alexa555 as a secondary antibody to detect V5-tagged NUP153.EcoDam fusion and EcoDam proteins (Supplementary Fig. 1c). DamID was performed as described[39] with few modifications. Three 16.7 mouse ES cell clones, expressing EcoDam (ED.B3) and NUP153.EcoDam fusion protein (NP.A2 and NP.D2) were used. Briefly, purified methyl PCR products were digested with DpnII to remove adapter sequences from the fragment ends and 30 ng of PCR products were treated as DNA templates to prepare paired-end Solexa libraries as previously described[68]. Genome Analyzer II (Illumina) was used to perform $2 \times 36$ cycles of paired-end sequencing. Sequencing reads from EcoDam overexpressing cells were used to normalize sequencing reads from NUP153.EcoDam overexpressing cells.

**Antibodies.** Anti-CTCF (1:1000; Millipore, 07-729), anti-SMC1A (1:1000: Bethyl, A300-055A), anti-SMC3 (1:1,000; Abcam, ab9263), anti-RAD21 (1:1000; Abcam, ab992), anti-FLAG (1:1000; Sigma-Aldrich, F1804), anti-NUP153 (1:1000; Abcam, ab24700), anti-GAPDH (1:10,000; Sigma-Aldrich, G9545), anti-Histone H3 (1:10,000; Abcam, ab1791), and anti-α-TUBULIN (1:1,000; Santa Cruz, sc-5286) were used in western blot analysis. Note that anti-NUP153 (Abcam, ab24700) can also detect NUP62 and was used to detect NUP62 by western blot analysis. Anti-Rpb1 NTD (3 μl, Cell Signaling, 14958), anti-CTCF (3 μl, Cell Signaling, 2899S), and anti-SMC3 (3 μg, Abcam, ab9263) were used in ChIP. Anti-LAMIN B1 (1:450; Abcam, ab16048), anti-V5 (1:400; Thermo Fisher Scientific, R960-25), anti-FLAG M2 (1:250; Sigma-Aldrich, F1804), anti-IgG(H+L)-Alexa555 (1:500; Thermo Fisher Scientific, A-21427), and anti-IgG(H+L)-Alexa488 (1:400; Thermo Fisher Scientific, A-11008 and A-32723) were used in immunofluorescence.

**Immunoprecipitation (IP) assay.** For IP assay, HEK293T cells that were transfected with FLAG-GFP or FLAG-NUP153 expression vector were lysed by sonication in IP lysis buffer (20 mM Tris–HCl, pH 7.9, 150 mM NaCl, 5 mM EDTA pH 8.0), 1% Nonident P-40, 10% glycerol, 1 mM phenylmethylsulfonyl fluoride (PMSF), 1 mM DTT, protease inhibitor cocktail (Sigma-Aldrich). After centrifugation, the supernatant was incubated with anti-FLAG M2 Affinity Gel beads (Sigma-Aldrich) at 4 °C for 2 h and the immune precipitates were subjected to western blotting. To prepare samples for the LC–MS/MS proteomics analysis, FLAG-NUP153 expression vector and mock transfected cells were lysed in elution buffer (10 mM PIPES, pH 6.8, 100 mM NaCl, 3 mM $MgCl_2$, 0.3 M sucrose, 0.5% Triton X-100, 1 mM PMSF, 1 mM DTT, protease inhibitor cocktail (Sigma-Aldrich)) for 10 min on ice, the nuclear fraction containing pellet was collected by centrifugation 3 min, $500 \times g$, 4 °C and was subjected to IP assay as described above. The immune precipitates were eluted by incubation with FLAG peptide (F4799) (Sigma-Aldrich) at room temperature for 15 min and were subjected to silver staining by using SilverXpress (Invitrogen) or utilized for LC–MS/MS proteomics analysis.

**LC-MS/MS proteomics analysis.** Samples in 1× Laemmli Sample buffer (Bio-Rad, 1610737) were run on a NuPAGE 4–12% Bis–Tris Protein gel (Invitrogen, NP0336PK2) in NuPAGE MES SDS Running Buffer (Invitrogen, NP0002) for ~5 min. The entire molecular weight range was excised and subjected to standardized in-gel trypsin digestion (http://www.genome.duke.edu/cores/proteomics/sample-preparation/documents/IngelDigestionProtocolrevised.pdf). Extracted peptides were lyophilized to dryness and resuspended in 12 μL of sample buffer (0.2% formic acid, 2% acetonitrile). Each sample was subjected to chromatographic separation on a nanoACQUITY UPLC (Waters) equipped with an ACQUITY UPLC BEH130 $C_{18}$ 1.7 μm, 75 μm I.D. × 250 mm column (Waters). The mobile phase consisted of (A) 0.1% formic acid in water and (B) 0.1% formic acid in acetonitrile. Following a 3 μL injection, peptides were trapped for 3 min on an ACQUITY UPLC M-Class Symmetry $C_{18}$ Trap Column 5 μm, 180 μm I.D. × 20 mm (Waters) at 5 μl/min in 99.9% A. The analytical column was then switched in-line and a linear elution gradient of 5% B to 40% B was performed over 30 min

at 400 nL/min. The analytical column was connected to a SilicaTip emitter (New Objective) with a 10 µm tip orifice and coupled to a Q Exactive Plus mass spectrometer (Thermo Fisher Scientific) through an electrospray interface operating in a data-dependent mode of acquisition. The instrument was set to acquire a precursor MS scan from $m/z$ 375–1600 at $R = 70,000$ (target AGC 1e6, max IT 60 ms) with MS/MS spectra acquired for the 10 most abundant precursor ions at $R = 17,500$ (target ABC 5e4, max IT 60 ms). For all experiments, HCD energy settings were 27 V and a 20 s dynamic exclusion was employed for previously fragmented precursor ions. Raw LC–MS/MS data files were processed in Proteome Discoverer (Thermo Fisher Scientific) and then submitted to independent Mascot search (Matrix Science) against a SwissProt database (*Human* taxonomy) containing both forward and reverse entries of each protein (20,322 forward entries). Search tolerances were 5 ppm for precursor ions and 0.02 Da for product ions using trypsin specificity with up to two missed cleavages. Carbamidomethylation (+57.0214 Da on C) was set as a fixed modification, whereas oxidation (+15.9949 Da on M) and deamidation (+0.98 Da on NQ) were considered dynamic mass modifications. All searched spectra were imported into Scaffold (v4.4, Proteome Software) and scoring thresholds were set to achieve a peptide false discovery rate of 1% using the PeptideProphet algorithm.

**Chromatin fractionation assay**. The chromatin fractionation assay was performed as previously described with minor modifications[35]. Briefly, HEK293T cells (~$4 \times 10^6$) were lysed in CSK buffer A (10 mM HEPES, pH 7.9, 10 mM KCl, 1.5 mM MgCl$_2$, 340 mM sucrose, 0.1% Triton X-100, 10% glycerol, 1 mM PMSF, 1 mM DTT, protease inhibitor cocktail (Sigma-Aldrich, P8340)) for 10 min on ice. Total cell lysate (T) was separated into supernatant (S1, containing cytoplasmic proteins) and nuclei by centrifugation 3 min, 500×$g$, 4 °C. Nuclei were further lysed in CSK buffer B (3 mM EDTA, 0.2 mM EGTA, pH 8.0, 1 mM DTT, 1 mM PMSF, protease inhibitor cocktail (Sigma-Aldrich, P8340)) and was incubated for 10 min on ice. The nuclear soluble fraction (S2, containing chromatin unbound proteins) and the nuclear insoluble fraction (P1) were separated by centrifugation 3 min, 500 × $g$, 4 °C. The P1 fraction was resuspended in MNase buffer (10 mM Tris–HCl, pH 7.9, 10 mM KCl, 3 mM CaCl$_2$, 300 mM sucrose, 1 mM PMSF, protease inhibitor cocktail (Sigma-Aldrich, P8340)) and chromatin was digested with 20 units of MNase (Thermo Fisher, EN0181) at room temperature for 15 min. The reaction was stopped by the addition of EGTA pH 8.0 at final concentration of 1 mM, followed by extraction with 250 mM ammonium sulfate at room temperature for 10 min. The chromatin-enriched fraction (S3, containing MNase-digested, chromatin-associated proteins) and the nuclear insoluble fraction (P2, containing insoluble, nuclear membrane and nuclear matrix proteins) were collected from the supernatant and the pellet, respectively, by centrifugation 5 min, 1700 × $g$, 4 °C.

**Total RNA extraction, reverse transcription, and real-time PCR**. Total RNA was extracted from cells using TRIzol reagent (Invitrogen) according to the manufacturer's instructions. For reverse transcription, cDNA was prepared using M-MLV Reverse Transcriptase (Thermo Fisher Scientific) with random hexamers (Sigma-Aldrich). Real-time PCR (qPCR) was performed using iTaq Universal SYBR Green Supermix (Bio-Rad) with specific primer sets indicated in Supplementary Data 10. Relative gene expression was calculated by the relative standard curve method. *GAPDH* expression was used to normalize data.

**IEG transcription induction in HeLa cells**. HeLa cells ($1 \times 10^6$) were transduced with the control (scramble) or human NUP153-specific shRNA lentivirus particles overnight at 37 °C followed by selection for 48 h in medium containing Puromycin (Puro, 2 µg/ml) (Sigma-Aldrich). To collect cells at the basal (minus EGF) IEG state, cells were pre-cultured in DMEM supplemented with 0.1% FBS (Sigma-Aldrich) for 24 h, followed by treatment with EGF (50 ng/ml) (Sigma-Aldrich, E9644) for 15, 30, 60, 90, and 120 min. For the rescue experiments, HeLa cells were transfected with control (scramble) or NUP153-specific shRNA vectors along with FLAG-hNUP153 expression vector using Xfect transfection reagent (Clontech) according to the manufacturer's instructions. At the 16 h time point, culture medium was replaced with Puro (2 µg/ml) containing medium and cells were incubated in this medium for 24 h, followed by incubation in Puro-free medium for another 24 h. To induce IEG transcription, cells were subjected to EGF treatment as described above.

**Chromatin immunoprecipitation (ChIP) assay**. ChIP experiments were performed as previously described[69]. Briefly, mouse ES cells (~$2 \times 10^6$) were cross-linked with 1% formaldehyde at room temperature for 10 min and the reaction was stopped by adding glycine (final concentration, 125 mM). Crosslinked cells were treated with a hypotonic buffer (10 mM HEPES–NaOH, pH 7.9, 1.5 mM MgCl$_2$, 10 mM KCl, 0.2% NP-40, 1 mM DTT, 1 mM PMSF, protease inhibitor cocktail (Sigma-Aldrich) at 4 °C for 10 min), and collected in a tube. The cells were lysed in ChIP lysis buffer (50 mM Tris–HCl, pH 7.9, 10 mM EDTA, 1% sodium dodecyl sulfate (SDS), 1 mM PMSF, protease inhibitor cocktail (Sigma-Aldrich) on ice for 10 min) and were subjected to sonication to shear the chromatin to 200–1000 bp-long DNA fragments. The lysate was diluted with 9× volumes of ChIP dilution buffer (16.7 mM Tris–HCl, pH 7.9, 167 mM NaCl, 1.2 mM EDTA, 1.1% Triton X-100, 1 mM PMSF), incubated with each antibody at 4 °C overnight with rotation,

and DNA–protein complexes were pulled down using Protein A/G agarose beads (Thermo Fisher, 20423). Agarose beads were washed with the following buffers: low-salt wash buffer (20 mM Tris–HCl, pH 7.9, 100 mM NaCl, 2 mM EDTA, pH 8.0, 0.1% SDS, 1% Triton X-100), high-salt wash buffer (20 mM Tris–HCl, pH 7.9, 500 mM NaCl, 2 mM EDTA, 0.1% SDS, 1% Triton X-100), and the LiCl buffer (10 mM Tris–HCl, pH 7.9, 250 mM LiCl, 1% NP-40, 1% deoxycholate acid, 1 mM EDTA, pH 8.0), and twice with TE buffer. DNA–protein complexes were eluted by incubating the beads in the elution buffer (1% SDS, 0.1 M NaHCO$_3$) for 15 min at room temperature with rotation. Reverse crosslinking was performed by incubating samples in 200 mM NaCl at 65 °C overnight followed by treatment with proteinase K (20 µg/ml) at 55 °C for 2 h. DNA was purified using MinElute PCR purification kit (QIAGEN). DNA samples were used to quantitate the occupancy of each protein using gene-specific primer sets (Supplementary Data 10) by real-time PCR.

**Immunostaining and DNA FISH**. For sequential LAMIN B1 immunostaining and *c-FOS* DNA FISH, HeLa cells ($5.5 \times 10^3$) were grown on 12-well glass slides (Invitrogen) overnight at 37 °C, and IEG transcription was induced as described above. Immunostaining was performed as previously described[67]. Briefly, fixed cells were subjected to immunostaining using anti-LAMIN B1 (Abcam, ab16048) antibody (1:450) at 4 °C overnight, washed three times in wash buffer (1× phosphate-buffered saline (PBS)/0.2% Tween-20 buffer) at room temperature for 5 min each, and incubated with goat polyclonal anti-IgG(H+L)-Alexa488 secondary antibody (1:500) for 1 h at room temperature. To remove excess secondary antibody, cells were washed three times in wash buffer for 5 min each. Slides were mounted using Vectashield mounting medium containing DAPI (Vector Labs). Slides were imaged using a Leica DM5500B microscope, and a Leica DFC365 FX CCD camera, image positions were recorded and slides were washed in 1× PBS/ 0.2% Tween 20 to remove the mounting medium and cells were re-fixed in 4% paraformaldehyde (Electron Microscopy Sciences) prior to DNA FISH experiment. To detect DNA signal at the *c-FOS* locus by FISH, BAC clone (RP11-293M10) (CHORI) was fluorescently labeled using Cy3-dUTP (ENZO) and nick translation kit (Sigma-Aldrich) according to the manufacturer's instructions. Human Cot-1 DNA (Thermo Fisher Scientific) (10 µg per 2 µg of nick translated BAC vector) was included into the reaction containing the nick translated vector to block the background DNA signal. The probe was precipitated by NaOAc-EtOH precipitation and the pellet was re-suspended in 50 µl of hybridization buffer (50% formamide, 2× saline sodium citrate (SSC), 2 mg/ml bovine serum albumin (Sigma-Aldrich), 10% dextran sulfate-500K (Millipore)) generating ~40 ng/µl labeled DNA probe. DNA FISH was performed as previously described[67]. Hybridization was performed using ~200 ng DNA probe per slide at 37 °C overnight in a humidified chamber. DNA FISH images at the recorded positions were obtained with a Leica DM5500B microscope, a Leica DFC365 FX CCD camera, and analyzed using ImageJ software (v.2.0.0). Distribution of *c-FOS* locus distance to nuclear periphery was measured in control and NUP153 KD HeLa cells at the indicated time points. Cumulative frequencies at a normalized distance (ND) of 0.0–0.12 are shown (Fig. 7). Frequency of *c-FOS* distribution at ND 0.0–0.45 is shown in Supplementary Fig. 7. ND = (*c-FOS* locus to periphery distance)/(cell diameter ($d$)), where $d = (2\times \text{nuclear area}/\pi)^{0.5}$. *$p < 0.05$; ***$p < 0.001$; the Kolmogorov–Smirnov (KS)-test was applied to calculate significance. To determine the cellular distribution of FLAG-NUP153 in HEK293T cells, FLAG-NUP153 transfected HEK293T cells ($5.5 \times 10^3$) were cultured on glass coverslips, and immunostaining was performed using anti-FLAG M2 (Sigma-Aldrich, F1804) antibody (1:250) as described above.

**Poly(A)$^+$ RNA FISH and alkaline phosphatase staining**. Poly(A)$^+$ RNA FISH was performed by using 5′ Cy3-labeled oligo-dT 50mer (Sigma-Aldrich) as previously described[46]. Briefly, hybridization was performed using 0.5 µg 5′ Cy3-labelled oligo-dT 50mer per slide at 37 °C in a humidified chamber overnight. Following hybridization, cells were washed twice for 15 min at 42 °C with 2× SSC, and once for 15 min at 42 °C in 0.5× SSC. Slides were mounted using mounting medium containing DAPI (Vector Labs) and cells were imaged by fluorescence microscopy. Alkaline phosphatase staining was performed using Red Alkaline Phosphatase Substrate kit (Vector Labs, SK-5100) according to the manufacturer's instructions. Bright-field images were taken using Leica EC3 color camera attached to Leica DM5500B microscope.

**Nuclear transport assay**. Hela cells were co-transfected with Rev-Glucocorticoid Receptor-GFP (RGG) expression vector (Gift from K. Ullman (University of Utah)) and control (scrambled) or hNUP153-specific shRNA vectors using Xfect reagent. Import and export assays were performed as previously described[54]. Briefly, for import assay, transfected HeLa cells were grown overnight on 12-well glass slides at 37 °C and treated with 250 nM dexamethasone (Dex) (Sigma-Aldrich, D4902) to induce RGG nuclear import for the indicated times. For the export assay, 120 min Dex-treated cells were washed with 1× PBS pH 7.2 and cultured in fresh culture medium for the indicated times. At the end of each time point, cells were fixed using 4% paraformaldehyde and mounted using DAPI containing mounting medium (Vector Labs). Images were obtained with a Leica DM5500B microscope, a Leica DFC365 FX CCD camera, and examined to calculate the percentage of cells with nuclear GFP-RGG signal.

**RNA-Seq**. Total RNA quality and concentration was assessed on a 2100 Bioanalyzer (Agilent Technologies) and Qubit 2.0 (Thermo Fisher Scientific), respectively. Total RNA (RIN value ≥ 8) from control and two NUP153 KD mouse ES cells were depleted of ribosomal RNA using the Illumina Ribo-zero Gold kit and converted into RNA-seq libraries using the Illumina Total RNA-seq kit. Libraries were indexed using a dual indexing approach allowing for multiple libraries to be pooled and sequenced on the same sequencing flow cell of an Illumina HiSeq 4000 sequencing platform. Before pooling and sequencing, fragment length distribution and library quality was first assessed on a Fragment Analyzer (Agilent). All libraries were pooled in equimolar ratio and sequenced. Libraries were sequenced at 50 bp single-end on the Illumina HiSeq 4000 instrument. About $110 \times 10^6$ reads per sample were generated. Once generated, sequence data was demultiplexed and Fastq files generated using Illumina's Bcl2Fastq v2 conversion software.

**ChIP-Seq**. ChIP DNA samples were quantified using the fluorometric quantitation Qubit 2.0 system (Thermo Fisher Scientific). ChIP-Seq libraries were prepared using the Roche Kapa BioSystem HyperPrep Library Kit to generate Illumina-compatible libraries. During adapter ligation, dual unique indexes were added to each sample. Resulting libraries were cleaned using SPRI beads and quantified using Qubit 2.0. Fragment length distribution of the final libraries was assessed on a Fragment Analyzer (Agilent). Libraries were then pooled into equimolar concentration and sequenced on an Illumina HiSeq 4000 instrument. Sequencing was done at 50 bp single-end and generated about $110 \times 10^6$ reads per sample. Sequence data was demultiplexed and Fastq files generated using Illumina's Bcl2Fastq v2 conversion software.

**RNA-Seq data analysis**. RNA-Seq reads were trimmed by Trim Galore (v.0.4.1, with -q 15) and then mapped with TopHat (v 2.1.1, with parameters --b2-very-sensitive --no-coverage-search and supplying the UCSC mm10 known gene annotation). The ERCC spike-in sequences were mapped separately. Gene-level read counts were obtained using the featureCounts (v1.6.1) by the reads with MAPQ greater than 30. Bioconductor package RUVseq (v 1.16.0) was used to normalize the read counts and edgeR (v 3.24.0) was employed for differential expression analysis. Fold change greater than 1.5 and false discovery rate (FDR) less than 0.05 was used to filter the significant differentially expressed genes.

**ChIP-Seq data analysis**. ChIP-Seq reads were trimmed by Trim Galore (0.4.1, with -q 15) and then mapped with bowtie2 (2.2.5, with parameters --very-sensitive) to mouse genome (UCSC mm10). The mapped reads were filtered by MAPQ greater than 30 by samtools (v 1.5) and duplicated reads were removed by picard (v 1.91). The peaks were called by MACS2 (v 2.1.0, with --pvalue 1e-5). The read coverages were quantified by the signal in reads per million per base pair https://github.com/BradnerLab/pipeline/blob/master/bamToGFF.py with parameters -m 500 -r -d. Metagene plots were used to display the average ChIP-seq signal across related regions of interest for enhancers and TSS separately. The average profile (metagene) was calculated by the mean of ChIP-seq signal profiles across the related regions of interest. For each metagene plot, the profile is displayed in rpm/bp in a ±2.5 kb or 5 kb region centered on the regions of interest. The number of enhancers or TSS were noted in the title of plots.

**DamID-Seq data analysis**. DamID-Seq reads were mapped with bowtie2 (2.2.5, with parameters --very-sensitive) to mouse genome (UCSC mm10). The mapped reads were filtered by MAPQ greater than 30 by samtools (v 1.5) and filtered by GATC at the 5′ ends. The peaks were called by MACS2 (v 2.1.0, with --q 0.05). To determine distribution of NUP153-DamID peaks across the genetic elements in mouse ES cells we used the following criterion. Promoters (−2 kb from TSS to +100 bp from TSS); GB (+100 bp from TSS to +1 kb from TTS); Intergenic sites (<−2 kb from TSS and >+1 kb from TTS). TSS, transcription start site; GB, gene body; TTS, transcription termination site.

**Definition of regulatory regions for the analyses of ChIP- and DamID-Seq data**. Several analyses in the manuscript rely on ChIP-or DamID-Seq analyses across different regulatory regions namely enhancers, promoters, and TAD boundaries. These regulatory regions were defined as follows. (A) Promoters were defined by gene start sites downloaded from UCSC Genome Browser goldenPath/mm10/database/knownGene. Active promoters were defined by the Fragments Per Kilobase of transcript per Million mapped reads (FPKM), which is calculated by cufflinks (v 2.1.1), greater than 1 in control RNA-seq. Inactive promoters were defined by FPKM no greater than 1. Chromatin structure at the transcriptionally active vs inactive TSS was validated using previously published H3K4me3 and H3K27me3 ChIP-Seq, respectively (GEO: GSE36905)[40]. (B) Enhancers were defined by utilizing the previously published ChIP-Seq data sets and determining the overlapping region of peaks with at least two enhancer-specific markers including CBP/P300 (GEO: GSE29184), H3K4me1 (GEO: GSE25409), or H3K27Ac (GEO: GSE42152). (C) TAD boundaries were defined by utilizing the previously published Hi-C data and TAD boundary coordinates reported[44]. (D) The overlap between NUP153 DamID peaks and CTCF or SMC3 ChIP peaks were defined using control (scramble shRNA) samples and were called by utilizing the

Bioconductor package ChIPpeakAnno (v. 3.19.5) with a maximal gap of 5 kb. The overlapping sites are referred to as co-occupied sites.

**Hi-C data analyses**. Mouse ES cell normalized 40 kb HiC Matrices (mm9) were downloaded from http://chromosome.sdsc.edu/mouse/hi-c/download.html. The Hi-C 2D map was plotted by R/Bioconductor package trackViewer (v. 1.23.2).

**Examining IEG chromatin structure in HeLa cells**. ENCODE HeLa-S3 ChIP-Seq data sets[55] for POL II (GEO: GSM733759), CTCF (GEO: GSM733785), RAD21 (GEO: GSM935571), CBP/P300 (GEO: GSM935553), H3K4me1 (GEO: GSM798322), H3K27Ac (GEO: GSM733684), and H3K4me3 (GEO: GSM733682) were utilized to examine chromatin structure across the *JUN* and *EGR1* genes (Supplementary Fig. 6b) using Human hg19 as a reference genome.

**Statistical analysis**. Quantitation of data was performed using the following statistical tests. Significance of the difference between control and knockdown cells for variables was analyzed with parametric Student's *t*-test. The nonparametric Kolmogorov–Smirnov (KS)-test was applied to calculate significance between the control and knockdown cells during the analyses of *c-FOS* locus spatial positioning with respect to nuclear periphery in a time course dependent manner.

**Reporting summary**. Further information on research design is available in the Nature Research Reporting Summary linked to this article.

## Data availability
Gene expression profiles, DamID-Seq and ChIP-Seq datasets have been deposited at GEO with accession code GSE135647. Proteomics data have been deposited to the ProteomeXchange Consortium via the PRIDE partner repository [https://www.ebi.ac.uk/pride/] with the Project ID PXD015441. The source data underlying Figs. 1a, b, d, 4a–d, 5, 6a–d, and 7a and Supplementary Figs. 1a, 2f, 5a, b, 6a, and 7 are provided as a Source Data file. All other relevant data supporting the key findings of this study are available within the article and its Supplementary Information files, and from the corresponding author upon reasonable request.

## Code availability
The custom analysis pipelines for all genomic analyses are available upon request with no restrictions.

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

## Acknowledgements

The authors are grateful to the members of the Yildirim lab for critical discussions and feedback. The authors thank L. Birnbaumer and S. Namekawa for critical reading and feedback on the manuscript. The authors acknowledge B. Van Steensel for the (V5)-EcoDam-pIND plasmid, K. Ullman for the REV-GFP-GR plasmid, C. Gersbach for the pMD2.G and psPAX2 plasmids, and J. Black for assistance in virus preparation. The authors thank E. Soderblom at the Duke Proteomics Shared Resource for assistance in the mass-spec experiment, N. Devos at the Duke Genome Sequencing Shared Resource for assistance in ChIP-, RNA-Seq library preparation and sequencing, the MGH Sequencing Core for assistance in the sequencing of DamID-Seq libraries, B. Cantor at the Duke Viral Shared Resource for assistance in virus preparation, and Duke Computing Cluster for computing support. This work was funded by the Whitehead Scholar Award (E.Y.) and

Duke University School of Medicine Department of Cell Biology start-up funds (E.Y) and in part by the NIH (R01-GM090278) (J.T.L.) and Howard Hughes Medical Institute (J.T.L.).

## Author contributions

E.Y. conceptualized and supervised the study. S.K. and E.Y. designed the experiments. S.K. performed all aspects of experiments associated with the identification of NUP153-interacting proteins, the role of NUP153 in regulation of the IEG transcription and chromatin structure in HeLa cells, and FLAG-NUP153 immunostaining, imaging and data analyses. Y.S. carried out all aspects of ChIP assays, Poly(A)$^+$ RNAFISH, and AP staining in mouse ES cells in collaboration with S.K. and E.Y. J.S. conducted and analyzed nuclear transport assays in collaboration with S.K. and E.Y. J.O. analyzed RNA-, DamID-, Hi-C, and ChIP-Seq data together with S.K. and E.Y. E.Y. generated Nup153-Dam expression plasmid, NUP153-Dam and Dam only mouse ES cell lines, and performed all aspects of DamID assay and DamID-Seq library preparation, LAMIN B1 immunostaining, c-FOS DNA-FISH, and related imaging and data analyses. The NIH and HHMI grants to J.T.L. provided funding for the generation of NUP153-(V5)-EcoDam-pIND vector, and DamID mouse ES cell clones, and sequencing of DamID-Seq libraries. S.K., J.O., Y.S., J.S., and E.Y. analyzed the experiments. S.K. and E.Y. wrote the paper with input from J.O., Y.S., and J.S.

## Competing interests

The authors declare no competing interests.
