## [Peer Review File · Nature Communications]

Reviewers' comments:

Reviewer #1 (Remarks to the Author):

The manuscript by Kadota et al. provides insights into the role of a nuclear pore component Nup153 in transcriptional and possibly architectural regulation of genes, likely via chromatin binding activity of Nup153. The authors present findings on the discovered interaction of Nup153 with cohesins and CTCF, and the effects of Nup153 depletion on chromatin targeting of these architectural proteins and of paused RNA pol II, with functional consequences on expression of select genes.

The most interesting and impactful finding in the paper is the observed loss of CTCF and cohesins on chromatin upon Nup153 KD, which the authors report both in mES cells genome-wide and in HeLa cells at selected inducible target genes. This is highly significant for the field since the findings suggest that nuclear pore proteins may regulate gene expression primarily through regulation of genome architecture and that recruitment of architectural proteins at specific loci, including enhancers, is regulated by Nups. The findings are also in line with previous reports implicating Nups in loop formation and enhancer function in yeast, flies and mammalian cells, but are the first (to my knowledge) to provide evidence that CTCF and cohesins actually depend on a Nup for proper chromatin targeting. Therefore, the paper would be of interest to a wide audience of scientists. However, the mechanistic implications of this need to be further addressed and validated, and the paper is particularly confusing in its analysis of genome-wide data, as discussed in specific points below:

Major points.

1. The comparison of CTCF, SMC3 and Nup153 ChIP-seq in +/- Nup153 depletion, along with RNA-seq, outlined in Figures 2-3, is difficult to interpret and requires revision. The grouping into the "WT specific", "common", and "153 KD specific" particularly does not make sense in light of Figure 2D:

1a) if the "153 KD specific" group consists of peaks of CTCF/SMC3 gained upon 153 loss, why do the mean CPM plots show the same or lower levels of CTCF or SMC3 in 153 KD_{1,2} for that group?

1b) Similarly, the "WT specific" and "common" groups also both show reduction of CTCF ChIP signal, but the "WT specific" group simply appears to have lower peaks generally. I understand that likely these groupings were made on the basis of peak calling, but they do not seem to provide any real differences in behavior of CTCF or SMC3 in response to Nup153 loss. Thus all other analysis based on these groupings does not seem to provide a lot of insight.

1c) It is also surprising, given the rest of the conclusions in the paper, that RNA-seq analysis shows that a large fraction of the Nup153-CTCF co-targets, in whichever grouping, show upregulation of expression upon 153 KD. Although Nup153 plays both repressive and activating roles in expression, from this paper and previously published work, the analysis does not give clear evidence to support the proposed role of Nup153 in promoting transcriptional initiation.

I suggest that authors simply define a subset of CTCF and of SMC3 sites that are reduced/lost upon 153 KD (such as the peaks shown in Figure 3B), using whatever reasonable threshold they choose. They can then compare whether these are more or less likely to be binding targets of Nup153, to provide further evidence that Nup153 functions in recruitment of CTCF/SMC3. Furthermore, these groupings (changed/unchanged, 153-bound/not bound) can then be compared to RNA-seq to define effects of Nup153/CTCF co-binding on transcription more clearly. In simplest way, how many genes behave like the example gene Calb2 in Figure 3B (bound by Nup153, reduced binding of CTCF and SMC3 in 153 KD, reduced expression in 153 KD)?

2. The first and second parts of the paper (the mES genome-wide and the HeLa IEG genes, respectively) are relatively disconnected – while the first part provides evidence that CTCF and cohesins may depend on Nup153 for binding (with some possible complex expression consequences), the second part shows that Nup153 is required for paused RNA pol II presence at poised IEG genes, which affects transcription initiation dynamics of these genes.

It thus remains possible that Nup153 functions primarily in stabilizing poised/paused RNA pol II binding (something that has been suggested for Nup98 in both yeast and HeLa cells), with all effects on architectural proteins being secondary. It would strengthen the main conclusions of the manuscript if the authors tested whether paused RNA pol II is affected by Nup153 KD in mES cells, particularly at loci such as *Calb2*, or bivalent genes, by ChIP-qPCR etc. Furthermore, can the authors find loci in mES cells where CTCF peaks are lost upon 153 KD, but RNA pol II is absent or unchanged? If they can, this would provide evidence that it is primarily CTCF or cohesins that are affected by Nup153.

3. The findings and the title here strongly implicate architectural/looping changes brought on by Nup153 KD, but the authors do not provide evidence for it directly. 3C-4C experiments at any of the shown loci would greatly strengthen these conclusions. Although I appreciate the difficulty of these experiments and they may not be absolutely required for publication, they would provide very strong and independent evidence that Nup153 loss results in architectural re-wiring of the genome.

Minor Points.

4. There appears to be no controls for the Nup153-FLAG construct used in Figure 1 and later on in the paper – does this construct express properly and most importantly, what is its nuclear distribution/can it incorporate into the NPC?

5. Generally, the insoluble nuclear pellet (Figure 1C-D) is not representative of just the NE/NPC, but contains a large amount of proteins that are thought to be part of the nuclear matrix. Insulator proteins such as CTCF have been previously shown to be part of this matrix fraction (for example Byrd K et al JCB 2003), thus presence of CTCF in the pellet should not be used as evidence for possible association with NPCs. If the authors do believe it is mainly representative of the nuclear membrane, they need to provide additional evidence such as western blotting for transmembrane proteins such as LBR and nuclear matrix proteins such as HnRNPU as negative control.

Reviewer #2 (Remarks to the Author):

In this study, the authors aim to investigate the role of NUP153 in the regulation of gene expression and in particular the molecular mechanism of how NUP153 association at regulatory regions (enhancers and TSS) impacts transcription.

They first used human cell lines to show interactions between NUP153 and the structural factors CTCF and cohesins via FLAG-tagged NUP153, IP-MS and nuclear fractionation. Then they switched to murine embryonic stem cells to investigate the role of NUP153 in the recruitment of CTCF and cohesin to regulatory elements. First they used DamID-NUP153 to map the binding sites across the genome and found that around 17.5% of these sites are located in enhancers. By comparing with ChIP-seq for CTCF and cohesin, they next found that almost half of the CTCF and Cohesin sites coincide with NUP153 sites. Then they performed a knock-down of NUP153, resulting in partial depletion to investigate further the link between NUP153 and CTCF/cohesin recruitment at key-regulatory sites (enhancers and TSS). The results could suggest a role of NUP153 in regulating specific developmental genes through the recruitment of CTCF and cohesin at enhancers and/or TSS. To dig further into the mechanisms of gene regulation by NUP153 they studied the induction by EGF of immediate early genes (IEGs) in WT versus NUP153 knock-down conditions. They show that IEG transcription initiation is mediated by NUP153 through its control on PolII occupancy

across TSSs. Further, knock-down of CTCF leads to the same deficiency, arguing that regulation of PolII occupancy by NUP153 may be directed through CTCF. Finally microscopy experiments show a correlation between NUP153 regulation of IEGs expression and their positioning close to the nuclear pore complex. Together these results led the authors to propose a model for NUP153 mediated chromatin conformation and transcription regulation at the IEG loci: NUP153-mediated recruitment of CTCF/cohesin at regulatory regions may stabilize PolII pausing at the nuclear pore complexes. Once transcription starts, NUP153-CTCF and cohesin are released, and IEG leaves the NPC vicinity.

This study brings a new mechanism for NUP153 role on gene regulation and how a nuclear compartment protein can interact with genome organizers to regulate gene expression. Although there are some concerns, this study is of great interest for the fields of nuclear organization, genome conformation, and gene regulation.

Here are my concerns:

- 1) The authors used a FLAG-NUP153, could they show the level of overexpression induced by this construction? Could it bring artefacts and biases in the study?
- 2) Why did the authors use first HEK293T cells and then HeLa cells for the IP and nuclear fractionation experiments? And why then moving the mouse ES cells?
- 3) In the DamID-NUP153, it would have been nice to get an example of the map of the binding sites (like in Fig 3). From Fig 3, it seems that NUP153 form focal sites, what is the average length of the sites? What is the profile of NUP 153 DamID compared to the one of LaminB1?
- 4) The authors show that 24352 NUP153 binding sites are located at TSSs. Which fraction of total TSSs does it represent?
- 5) 17.5% of the enhancers are bound by NUP153. Do these enhancers share specific features? Are they related to the TSSs bound by NUP153?
- 6) How many TSSs and/or enhancers are bound by NUP153, CTCF and cohesin at the same time?
- 7) The authors showed that on average CTCF/cohesin sites are located at 5kb from a NUP153 site. What is the biological relevance of being that far?
- 8) What is the percentage of NUP153 sites occupied by CTCF and cohesin? At promoters, enhancers? May be interesting to get a summary of this (a graph?). If the function of NUP153 in gene expression is mediated through CTCF and cohesin, we could imagine that most of the NUP153 sites are occupied by CTCF and cohesin, whereas the other way around may not be necessary (as CTCF and cohesin have other functions).
- 9) The NUP153 knock-down shows a 60% drop in NUP153 expression. This is a partial downregulation. Enough to assess the roles of NUP153? Why not using the DEGRON system for example?
- 10) It would have been interesting to perform a DamID-NUP153 in the KD to assess which sites are staying, and to compare with the changes in CTCF/cohesin peaks.
- 11) In Fig 2C: the 2 replicates for the KD give quite different results in terms of peaks. How many peaks are in common? Could the authors comment on this? What kind of sites is more variable (ie promoters, enhancers, gene body, intergenic)?
- 12) In Fig 2D, it is not clear to me why there is a decrease in CTCF or cohesin for common or NUP-KD specific sites: the common sites should be there in both conditions, and the NUP-KD specific should be there only in the mutant cells. Am I missing something there? (this comment goes for the other parts of the manuscript referring to the 3 groups of sites).
- 13) The authors state "NUP153 mediates TSSs and enhancer specific binding of CTCF and cohesin". However, they show that a lot of these sites are NUP153 independent (common sites) or even present only in absence of NUP153 (NUP-KD specific sites). So is there a specific type of regulatory elements bound by CTCF / cohesin in a relevant NUP153-dependent manner?
- 14) In the list of genes differentially regulated in NUP153-KD cells, how many are bound by NUP153 directly? In their promoter or enhancer?
- 15) In Fig 3B:
 - is there a transcription change for Rtn4rl1?
 - it seems on the figure that cohesin sites, and their change after KD, always follow CTCF kinetics. Is it always the case?

- 16) In Fig 3C, the 2 critical CTCF sites (asterisks) in the HoxA cluster are not related to a NUP153 peak. Do the authors have an hypothesis?
- 17) In Fig 7A, what is the significance of the change in the distance between c-fos and the nuclear periphery?
- 18) In general, the results are interesting in a role of NUP153 in stabilizing PolIII occupancy probably through CTCF/cohesin recruitment. However, levels/numbers of changes in the KD experiments are most of the time quite minor. However this may be explained by the partial downregulation via the knock-down. A Degron system may give more significant changes.

Reviewer #3 (Remarks to the Author):

The manuscript entitled "Nucleoporin 153 links nuclear pore complex to chromatin architecture by mediating CTCF and cohesin binding at cis-regulatory elements" by Dr Yildirim and colleagues describe the interaction of Nucleoporin 153 with potential interaction partners that are part of the chromatin architecture.

This reviewer is mainly assessing the proteomics part of the manuscript.

Comments:

The authors are not describing the pull down experiment that well in the manuscript, especially not the database identification and quantitation. It should be described so that it could be repeated by others. As far as I understand the authors are linking the NUP153 to a FLAG tag and then use anti-FLAG affinity beads for the pull down. The control is just the beads themselves. Should the control not have been the beads with FLAG tag on, as proteins will bind to the FLAG tag? The eluted proteins are then separated on a gel and in-gel digestion is performed. The peptides are analyzed by LC-MSMS using a Q-exactive plus instrument. The database searching was performed in PD, but there is no information on how the authors have identified the true interaction partners? And according to the manuscript the experiment was performed once? IP and protein identification is associated with a lot of errors and pull down experiments should be performed at least 3 times in order to say anything quantitative about protein identifications, especially when the peptides comes from in-gel digestion.

This reviewer tried for several days to download the raw data files from ProteomeXchange using the Username and password provided but it was not possible. So it is not possible to evaluate the protein identifications or quantitation.

RESPONSE TO REVIEWERS

We would like to thank all three Reviewers for the constructive comments, and the positive response we received on our manuscript by Kadota et al. entitled "*Nucleoporin 153 links nuclear pore complex to chromatin architecture by mediating CTCF and cohesin binding at cis-regulatory elements*" (NCOMMS-19-31033). We have revised the manuscript according to the Reviewers' critiques. Please find our response below following the Reviewer's boldfaced comments. We believe these changes have improved the manuscript considerably.

Reviewer #1 (Remarks to the Author):

The manuscript by Kadota et al. provides insights into the role of a nuclear pore component Nup153 in transcriptional and possibly architectural regulation of genes, likely via chromatin binding activity of Nup153. The authors present findings on the discovered interaction of Nup153 with cohesins and CTCF, and the effects of Nup153 depletion on chromatin targeting of these architectural proteins and of paused RNA pol II, with functional consequences on expression of select genes.

The most interesting and impactful finding in the paper is the observed loss of CTCF and cohesins on chromatin upon Nup153 KD, which the authors report both in mES cells genome-wide and in HeLa cells at selected inducible target genes. This is highly significant for the field since the findings suggest that nuclear pore proteins may regulate gene expression primarily through regulation of genome architecture and that recruitment of architectural proteins at specific loci, including enhancers, is regulated by Nups. The findings are also in line with previous reports implicating Nups in loop formation and enhancer function in yeast, flies and mammalian cells, but are the first (to my knowledge) to provide evidence that CTCF and cohesins actually depend on a Nup for proper chromatin targeting. Therefore, the paper would be of interest to a wide audience of scientists. However, the mechanistic implications of this need to be further addressed and validated, and the paper is particularly confusing in its analysis of genome-wide data, as discussed in specific points below:

We are very pleased that Reviewer #1 found our study to be interesting, impactful, and our findings to be highly significant for the field, and to be an interest to a wide audience

of scientists. We also thank the Reviewer for his/her excellent feedback and constructive comments. We revised the manuscript according to the Reviewer's recommendations. Please find our response below following the Reviewer's boldfaced comments.

Major points.

1. The comparison of CTCF, SMC3 and Nup153 ChIP-seq in +/- Nup153 depletion, along with RNA-seq, outlined in Figures 2-3, is difficult to interpret and requires revision. The grouping into the "WT specific", "common", and "153 KD specific" particularly does not make sense in light of

1a) if the "153 KD specific" group consists of peaks of CTCF/SMC3 gained upon 153 loss, why do the mean CPM plots show the same or lower levels of CTCF or SMC3 in 153 KD1,2 for that group?

1b) Similarly, the "WT specific" and "common" groups also both show reduction of CTCF ChIP signal, but the "WT specific" group simply appears to have lower peaks generally. I understand that likely these groupings were made on the basis of peak calling, but they do not seem to provide any real differences in behavior of CTCF or SMC3 in response to Nup153 loss. Thus all other analysis based on these groupings does not seem to provide a lot of insight.

1c) It is also surprising, given the rest of the conclusions in the paper, that RNA-seq analysis shows that a large fraction of the Nup153-CTCF co-targets, in whichever grouping, show upregulation of expression upon 153 KD. Although Nup153 plays both repressive and activating roles in expression, from this paper and previously published work, the analysis does not give clear evidence to support the proposed role of Nup153 in promoting transcriptional initiation.

I suggest that authors simply define a subset of CTCF and of SMC3 sites that are reduced/lost upon 153 KD (such as the peaks shown in Figure 3B), using whatever reasonable threshold they choose. They can then compare whether these are more or less likely to be binding targets of Nup153, to provide further evidence that Nup153 functions in recruitment of CTCF/SMC3. Furthermore, these groupings (changed/unchanged, 153-bound/not bound) can then be compared to RNA-seq to define effects of Nup153/CTCF co-binding on transcription more clearly. In simplest way, how many genes behave like the example gene Calb2 in Figure 3B (bound by

Nup153, reduced binding of CTCF and SMC3 in 153 KD, reduced expression in 153 KD)?

We thank the reviewer for her/his insightful comments and suggestions. We followed the Reviewer's recommendations and revised mouse ES cell genome-wide analysis (Figure 2, 3, S2 and S3) as follows:

1. We first identified NUP153-positive TSS and enhancers (Figure 2B). Next, we added a new dataset by utilizing previously published mouse ES cell Hi-C data (Dixon et al. 2012) and determined NUP153-positive TAD boundaries (Figure 2B). We identified 31.5% TSS, 17.5% enhancers and 66.9% TAD boundaries to contain NUP153 binding sites (Figure 2B and Table S2).

2. We compared CTCF and SMC3 binding sites between control and NUP153 KD ES cells and found significant loss of CTCF (~60%) and cohesin (~86%) binding in NUP153 deficient ES cells (Figure 2C).

3. Given that cohesin binding relies on CTCF (Parelho et al. 2008; Merkenschlager et al. 2013), we focused on CTCF binding sites in control cells. To determine if NUP153 impacts CTCF and cohesin binding selectively at a genetic element, we identified CTCF-positive TSS, enhancers and TAD boundaries and compared CTCF or SMC3 binding across these sites. We found significant decrease in CTCF and SMC3 binding across all three genetic elements in NUP153 deficient cells (Figure 2D).

4. Per Reviewer's suggestion, we grouped CTCF-positive sites to investigate genes that might be co-regulated by NUP153 and CTCF. To do so, we calculated the mean CTCF binding in NUP153 deficient cells in comparison to control cells which allowed us to group CTCF-positive sites into two Groups: Group I, contained CTCF sites that showed greater mean CTCF binding in control cells over NUP153 deficient cells. Group II, contained CTCF sites that showed equal or lesser CTCF binding in control cells over NUP153 deficient cells (top, Figure 2E). Group I TSS sites constituted ~10% (1,123/11,726) of the total CTCF binding sites and half of these sites (~5%, 558/11,726) were NUP153 positive. In line with this finding, metagene profiles across TSS, enhancer and TAD boundaries showed higher NUP153 binding at Group I sites over Group II sites (Figure 2E, Table S5). This data suggested that degree of NUP153 binding correlates with differential change in CTCF binding at each genetic element. Based on these findings, we concluded that

NUP153 mediates CTCF and cohesin binding at TSS, enhancer and TAD-boundaries. This raises the possibility that NUP153 may be critical for enhancer-promoter functions or chromatin organization functions of CTCF and cohesin during gene expression.

5. We next investigated how NUP153-dependent changes in CTCF binding may influence transcription. We found that 34.4% (245/711, 34.4%) of the total differentially regulated genes associated with CTCF-positive TSS (Table S8). Within this group, 19.4% (138/711) associated with Group I CTCF TSS sites and 7% (47/711) were NUP153-positive CTCF-Group I genes (Table S8).

6. Genes *Rtn4r11* and *Calb2* belong to NUP153-positive CTCF-Group I genes and we showed representative NUP153 DamID-Seq, CTCF, SMC3 ChIP-Seq and RNA-Seq tracks for these genes in Figure 3C.

2. The first and second parts of the paper (the mES genome-wide and the HeLa IEG genes, respectively) are relatively disconnected – while the first part provides evidence that CTCF and cohesins may depend on Nup153 for binding (with some possible complex expression consequences), the second part shows that Nup153 is required for paused RNA pol II presence at poised IEG genes, which affects transcription initiation dynamics of these genes.

It thus remains possible that Nup153 functions primarily in stabilizing poised/paused RNA pol II binding (something that has been suggested for Nup98 in both yeast and HeLa cells), with all effects on architectural proteins being secondary. It would strengthen the main conclusions of the manuscript if the authors tested whether paused RNA pol II is affected by Nup153 KD in mES cells, particularly at loci such as *Calb2*, or bivalent genes, by ChIP-qPCR etc. Furthermore, can the authors find loci in mES cells where CTCF peaks are lost upon 153 KD, but RNA pol II is absent or unchanged? If they can, this would provide evidence that it is primarily CTCF or cohesins that are affected by Nup153.

We thank the Reviewer's suggestion on investigating POL II binding in NUP153 deficient ES cells. We agree with the Reviewer that analyzing POL II dynamics in NUP153 deficient cells would be informative. However, based on our findings, we concluded that altered POL II binding at IEGs is secondary to altered CTCF occupancy at IEG enhancer sites in NUP153 deficient cells. Our conclusion is based on the following assays and results. By

utilizing IEG loci in HeLa cells, we demonstrated that NUP153 depletion alters IEG transcription (Figure 4B) coupled with altered POL II occupancy at TSS (Figure 4D) and altered CTCF binding across enhancers (Figure 5). By depleting CTCF in HeLa cells, we further provide evidence that CTCF depletion mimics NUP153 knockdown phenotype resulting in altered POL II binding at TSS of IEGs (Figure 6C). Importantly, depletion of both NUP153 and CTCF in HeLa cells did not result in an additive effect in suppression of IEG transcription. This finding suggested that NUP153 and CTCF regulate IEG transcription through the same regulatory mechanism. In the light of these findings, we hope that the Reviewer agrees that existing data is sufficient to support a mechanism through which NUP153 and CTCF co-regulates POL II binding and transcription at selective genes, such as IEGs, and differentially regulated NUP153 positive CTCF Group I genes in ES cells. Furthermore, we believe that the two parts (ES cell and HeLa) of the manuscript are better connected in the revised manuscript due to the reanalyzes of the genome-wide ES cell data. We respectfully would like to suggest that NUP153-dependent changes in POL II dynamics is a focus of a future study.

3. The findings and the title here strongly implicate architectural/looping changes brought on by Nup153 KD, but the authors do not provide evidence for it directly. 3C-4C experiments at any of the shown loci would greatly strengthen these conclusions. Although I appreciate the difficulty of these experiments and they may not be absolutely required for publication, they would provide very strong and independent evidence that Nup153 loss results in architectural re-wiring of the genome.

We agree with the Reviewer that including 3C or 4C data for the IEGs would have strengthened the conclusions of our study. However, as the Reviewer rightfully acknowledged, these assays are complex and require ample time to be standardized. Due to the time limitation of this revision, we unfortunately are unable to provide 3C or 4C data. Despite this fact, two reports published during the preparation of this manuscript support our conclusions. These reports provided 3C evidence that CTCF-dependent enhancer-promoter looping is critical for IEG (e.g. EGR1) transcription (Zheng et al. 2019; Sekiya et al. 2019). Our findings revealed that NUP153 depletion alters CTCF binding that is critical for these interactions suggesting that NUP153 acts as a regulator of chromatin organization by mediating CTCF binding. Furthermore, in the revised manuscript, we utilized previously published Hi-C data (Dixon et al. 2012) and provided new data showing that ES cell genome contains 66.9% NUP153-positive TAD boundaries (Figure 2B, Table

S2C). Notably, we detect significant decrease in CTCF and cohesin binding at CTCF-positive TAD boundaries. These data strengthen our findings for a potential role of NUP153 in CTCF binding at TAD boundaries and mediating chromatin architecture.

Minor Points.

4. There appears to be no controls for the Nup153-FLAG construct used in Figure 1 and later on in the paper – does this construct express properly and most importantly, what is its nuclear distribution/can it incorporate into the NPC?

In the revised manuscript, we included western blot data showing proper expression of FLAG-NUP153 fusion protein in HEK293T cells (Figure S1A). We also show that NUP153-FLAG fusion protein localizes to the nuclear periphery and slightly to the nucleoplasm similar to the nuclear distribution of the endogenous NUP153 (Figure S1B). Furthermore, mass-spec analysis has revealed that NUP153-FLAG interacts with the known NUP153-interacting proteins and the NPC components (page 7) suggesting that NUP153-FLAG fusion protein behaves similarly to the endogenous NUP153 protein.

5. Generally, the insoluble nuclear pellet (Figure 1C-D) is not representative of just the NE/NPC, but contains a large amount of proteins that are thought to be part of the nuclear matrix. Insulator proteins such as CTCF have been previously shown to be part of this matrix fraction (for example Byrd K et al JCB 2003), thus presence of CTCF in the pellet should not be used as evidence for possible association with NPCs. If the authors do believe it is mainly representative of the nuclear membrane, they need to provide additional evidence such as western blotting for transmembrane proteins such LBR and nuclear matrix proteins such as HnRNPU as negative control.

We revised the manuscript accordingly. We included the fact that the P2 insoluble nuclear fraction contains the nuclear matrix in addition to the nuclear membrane (NE/NPC) and cited the related reference (page 7).

Reviewer #2 (Remarks to the Author):

In this study, the authors aim to investigate the role of NUP153 in the regulation of gene expression and in particular the molecular mechanism of how NUP153 association at regulatory regions (enhancers and TSS) impacts transcription. They

first used human cell lines to show interactions between NUP153 and the structural factors CTCF and cohesins via FLAG-tagged NUP153, IP-MS and nuclear fractionation. Then they switched to murine embryonic stem cells to investigate the role of NUP153 in the recruitment of CTCF and cohesin to regulatory elements. First they used DamID-NUP153 to map the binding sites across the genome and found that around 17.5% of these sites are located in enhancers. By comparing with ChIP-seq for CTCF and cohesin, they next found that almost half of the CTCF and Cohesin sites coincide with NUP153 sites. Then they performed a knock-down of NUP153, resulting in partial depletion to investigate further the link between NUP153 and CTCF/cohesin recruitment at key-regulatory sites (enhancers and TSS). The results could suggest a role of NUP153 in regulating specific developmental genes through the recruitment of CTCF and cohesin at enhancers and/or TSS. To dig further into the mechanisms of gene regulation by NUP153 they studied the induction by EGF of immediate early genes (IEGs) in WT versus NUP153 knock-down conditions. They show that IEG transcription initiation is mediated by NUP153 through its control on PolII occupancy across TSSs. Further, knock-down of CTCF leads to the same deficiency, arguing that regulation of PolII occupancy by NUP153 may be directed through CTCF. Finally microscopy experiments show a correlation between NUP153 regulation of IEGs expression and their positioning close to the nuclear pore complex. Together these results led the authors to propose a model for NUP153 mediated chromatin conformation and transcription regulation at the IEG loci: NUP153-mediated recruitment of CTCF/cohesin at regulatory regions may stabilize PolII pausing at the nuclear pore complexes. Once transcription starts, NUP153-CTCF and cohesin are released, and IEG leaves the NPC vicinity. This study brings a new mechanism for NUP153 role on gene regulation and how a nuclear compartment protein can interact with genome organizers to regulate gene expression. Although there are some concerns, this study is off great interest for the fields of nuclear organization, genome conformation, and gene regulation.

We are very pleased that Reviewer #2 found our study to be off great interest for the fields of nuclear organization, genome conformation and gene regulation. We also thank the Reviewer for his/her excellent feedback and constructive comments. We revised the manuscript according to the Reviewer's recommendations. Please find our response below following the Reviewer's boldfaced comments.

Here are my concerns:

1) The authors used a FLAG-NUP153, could they show the level of overexpression induced by this construction? Could it bring artefacts and biases in the study?

In the revised manuscript, we included western blot data showing proper expression of FLAG-NUP153 fusion protein in HEK293T cells (Figure S1A). We also showed that FLAG-NUP153 fusion protein localizes to the nuclear periphery and slightly into the nucleoplasm similar to the nuclear distribution of the endogenous NUP153 (Figure S1B). In addition to these data sets, our mass-spec analysis has revealed that FLAG-NUP153 interacts with the known NUP153-interacting proteins and the NPC components (page 7). Lastly, we used FLAG-NUP153 construct to restore NUP153 levels in NUP153 deficient HeLa cells in which we detect altered IEG transcription. Exogenous expression of FLAG-NUP153 in NUP153 deficient cells resulted in rescue of IEG transcription (Figure 4C). These data suggest that FLAG-NUP153 fusion protein behaves similarly to the endogenous NUP153 protein.

2) Why did the authors use first HEK293T cells and then HeLa cells for the IP and nuclear fractionation experiments? And why then moving the mouse ES cells?

1. We chose HEK293T cells for the following reasons: In order to define regulatory role of NUP153 in gene regulation, we devised an unbiased proteomics screen using NUP153-FLAG fusion protein as a bait in an affinity purification assay. This assay relies on a high transfection efficiency cell system such as HEK293T to achieve efficient pull down. We were successful in efficiently and properly expressing NUP153-FLAG fusion protein in HEK293T cells (Figure S1A-B) and carried out mass-spec analyses that revealed NUP153 interaction with cohesin subunits (Figure 1A).

2. We chose HeLa cells for the chromatin fractionation assay for following reason: HeLa cells have been successfully and widely utilized for the chromatin fractionation assays as the protein markers for each fraction can be detected clearly. Our protocol (Figure 1C) was based on the assay developed by Wysocka et al., 2001 using HeLa cells and allowed us to detect various fraction markers and NUP153 in the chromatin fraction (Figure 1D).

3. We chose ES cells for addressing gene regulatory role of NUP153 for following reasons: My lab is interested in determining cell-type specific gene regulatory function of NUP proteins during development. Pluripotent mouse ES cells provide an excellent model to study gene regulatory events during early development. Upon identifying CTCF and

cohesin as NUP153 interacting proteins using HEK293T and HeLa cells (Figure 1, S1A-B), we used mouse ES cells to first map NUP153 binding sites by DamID-Seq (Figure 2A-B, Table S1). This was followed by targeting NUP153 in mouse ES cells to investigate its role in regulation of CTCF and cohesin binding (Figure 2) and transcription (Figure 3). Our findings showed that NUP153 is a critical regulator of CTCF and cohesin binding at cis-regulatory elements and TAD boundaries. Mouse ES cell genome is well-characterized to contain bivalent genes. Thus, mouse ES cells also allowed us to study how NUP153 influence bivalent gene expression and found that a significant number of bivalent genes are differentially regulated in NUP153 deficient ES cells (Figure 3A and Table S9).

3) In the DamID-NUP153, it would have been nice to get an example of the map of the binding sites (like in Fig 3). From Fig 3, it seems that NUP153 form focal sites, what is the average length of the sites? What is the profile of NUP 153 DamID compared to the one of LaminB1?

1. In Figure 3 and S4, we provide NUP153 DamID-Seq tracks showing NUP153 binding. By utilizing metagene profiles, we also present mean NUP153 binding profiles across various genetic elements including TSS, enhancers and TAD boundaries (Figure 2B, S2A-C, S3B).

2. Based on the NUP153 binding sites (Table S1), we calculated the average length of NUP153 binding sites as ~285 base pairs.

3. We thank the reviewer for the suggestion to compare Lamin B1 interaction profile to NUP153 profile. Lamin B1 interaction sites, referred to lamina-associated domains (LADs) were mapped by Peric-Hupkes et al. (Peric-Hupkes et al. 2010) and were shown to range in size from ~40 kb to 15 Mb. We believe that comparing the LADs to NUP153 interaction sites is important. However, we think that this analysis does not fit with the scope of the current study. We hope that the Reviewer agrees with us and the fact that this analysis should be included in a future follow up study that focuses on understanding how chromatin-nuclear envelope interactions are distributed among the NPCs and the lamina.

4) The authors show that 24352 NUP153 binding sites are located at TSSs. Which fraction of total TSSs does it represent?

We apologize for the confusion in the text and presentation of Figure 2 which lead to the notion that we identified 24,352 NUP153 binding sites at TSS. We revised the text and Figures 2B accordingly. In the revised manuscript, we included metagene profiles showing NUP153-positive and NUP153-negative TSS, enhancer and Topologically Associating Domain (TAD) boundaries (Figure 2B). In Figure 2B, we also included a Table showing the number and percent of NUP153-positive and NUP153-negative TSS, enhancer and TAD boundaries. Specifically, out of the 24,352 total TSS we identified 31.5% NUP153-positive TSS (7721/24,352).

5) 17.5% of the enhancers are bound by NUP153. Do these enhancers share specific features? Are they related to the TSSs bound by NUP153?

1. To determine if NUP153-positive enhancers share specific features, we utilized the previously published ChIP-Seq data for enhancer specific histone modifications (H3K4me1 and H3K27Ac) and P300/CBP and performed metagene profiling across NUP153-positive and NUP153-negative enhancers (+/- 2.5 kb) (Figure S2D). We found that NUP153-positive enhancers exhibited higher binding for all the marks that we have examined.

2. A comprehensive analyses including Hi-C should be carried out to determine promoter-enhancer associations, and examine these interactions in the context of NUP153 binding. While we appreciate the Reviewer for this question, we believe this analysis is beyond the scope and the timeline of the current study and we hope to address it in the future.

6) How many TSSs and/or enhancers are bound by NUP153, CTCF and cohesin at the same time?

We revised Table S3 and included number of CTCF sites which show cohesin and NUP153 co-occupancy at TSS and enhancers. We found that 179 (10.4%) TSS- and 193 (13.9%) enhancer-specific CTCF sites also contain cohesin and NUP153. We included this information in the text (page 9).

7) The authors showed that on average CTCF/cohesin sites are located at 5kb from a NUP153 site. What is the biological relevance of being that far?

As discussed in the text (page 8), we utilized DamID-Seq to map NUP153 interaction sites. DamID relies on NUP153-Dam fusion protein to associate with the GATC sequence that

resides near a NUP153 binding site, resulting in adenine N⁶ methylation. Furthermore, Dam-fusion proteins can interact and catalyze adenine N⁶-methylation up to ~5 kb distance (van Steensel and Henikoff. 2000). Thus, through this method, we can detect both transient and stable NUP153 interaction sites. We aimed to capture as many CTCF or SMC3 NUP153 interaction sites as possible and thus calculated average distance between the proteins (Figure S2E). We unfortunately do not have a direct answer so as to what might be the biological relevance of ~5 kb distance. Using this distance as a criterion, we aimed to detect NUP153/CTCF and NUP153/SMC3 sites that can functionally cooperate during gene regulation.

8) What is the percentage of NUP153 sites occupied by CTCF and cohesin? At promoters, enhancers? May be interesting to get a summary of this (a graph?). If the function of NUP153 in gene expression is mediated through CTCF and cohesin, we could imagine that most of the NUP153 sites are occupied by CTCF and cohesin, whereas the other way around may not be necessary (as CTCF and cohesin have other functions).

We listed the number and the percentage of sites that show overlap between NUP153 and CTCF or SMC3, and association of the sites with TSS or enhancers in Table S3.

9) The NUP153 knock-down shows a 60% drop in NUP153 expression. This is a partial downregulation. Enough to assess the roles of NUP153? Why not using the DEGRON system for example?

We appreciate the Reviewer's suggestion on utilizing the Auxin-inducible degron (AID) system to study NUP153 function. We agree that the AID system would have been superior to the other targeting technologies including shRNA which we utilized in the current study. However, we can detect clear and consistent phenotype by knocking down NUP153 using two different shRNAs in two different cell systems (HeLa and mouse ES cells). Importantly, we can rescue the phenotype by restoring NUP153 levels (Figure 4C) suggesting that it is NUP153 specific. We hope that the Reviewer agrees with us that the use of AID system would require repeating of all the experiments and related analyses which would be timely. We think that the AID system could be utilized for future studies.

10) It would have been interesting to perform a DamID-NUP153 in the KD to assess which sites are staying, and to compare with the changes in CTCF/cohesin peaks.

DamID method relies on the catalytic activity of the Dam enzyme which methylates N⁶ adenine within the GATC sequence at the Dam-fusion protein interaction sites. Due to this fact, Dam-NUP153 ES cells would not be a reliable cell line to knockdown NUP153 and examine which sites remain, as majority of the N⁶ adenine methylation would likely be persistent. In the second half of our manuscript, we studied NUP153 binding dynamics in relation to CTCF and cohesin binding in control and NUP153 KD HeLa cells across the IEG loci (Figure 5). We found that NUP153 is critical for CTCF and cohesin binding at the enhancers and promoters.

11) In Fig 2C: the 2 replicates for the KD give quite different results in terms of peaks. How many peaks are in common? Could the authors comment on this? What kind of sites is more variable (ie promoters, enhancers, gene body, intergenic)?

As shown in Figure 2C, we detected 1,812 CTCF and 81 SMC3 sites that are shared between the two NUP153 KD ES cell lines. Examining CTCF and SMC3 binding across different genetic elements including promoter, intergenic sites and gene bodies revealed that distribution of CTCF and SMC3 binding at these elements were comparable between the two NUP153 KD cell lines (Figure S3A). This information was included in the manuscript.

12) In Fig 2D, it is not clear to me why there is a decrease in CTCF or cohesin for common or NUP-KD specific sites: the common sites should be there in both conditions, and the NUP-KD specific should be there only in the mutant cells. Am I missing something there? (this comment goes for the other parts of the manuscript referring the the 3 groups of sites).

We apologize the Reviewer for the confusion in how the data was presented causing misunderstanding. We revised mouse genome-wide analyses and related Figures (Figure 2, 3, S2, S3). In Figure 2D, we presented CTCF and SMC3 ChIP-Seq data by focusing on the CTCF-positive sites (n=11,726) in control ES cells. To determine if NUP153 influences CTCF binding at a specific genetic element(s), we sub-grouped CTCF binding sites into TSS (n=2,164) and enhancer binding sites (n=2,272). We found that CTCF and SMC3 binding was significantly altered in NUP153 deficient cells at CTCF positive TSS and enhancers. To provide more insights on whether NUP153 effects CTCF or cohesin binding at TAD boundaries, we further mapped TAD boundaries by utilizing previously published

Hi-C data (Dixon et al. 2012). We next identified CTCF-positive TAD boundaries (n=2,238), which showed significant decrease in CTCF or SMC3 binding in NUP153 KD ES cells.

13) The authors state” NUP153 mediates TSSs and enhancer specific binding of CTCF and cohesin”. However, they show that a lot of these sites are NUP153 independent (common sites) or even present only in absence of NUP153 (NUP-KD specific sites). So is there a specific type of regulatory elements bound by CTCF / cohesin in a relevant NUP153-dependent manner?

We apologize the Reviewer for the confusion in how the data was presented causing misunderstanding. We revised the mouse genome-wide analyses and related Figures (Figure 2, 3, S2, S3). In Figure 2D, we presented CTCF and SMC3 ChIP-Seq data by focusing on the CTCF-positive sites (n=11,726) in control ES cells. To determine if NUP153 influences CTCF binding at a specific genetic element(s), we sub-grouped CTCF binding sites into TSS (n=2,164) and enhancer binding sites (n=2,272). We found that CTCF and SMC3 binding was significantly altered in NUP153 deficient cells at CTCF positive TSS and enhancers. To provide more insights on whether NUP153 effects CTCF or cohesin binding at TAD boundaries, we further mapped TAD boundaries by utilizing previously published Hi-C data (Dixon et al. 2012). We next identified CTCF-positive TAD boundaries (n=2,238), which showed significant decrease in CTCF or SMC3 binding in NUP153 KD ES cells. As shown in Figure 2E, CTCF positive Group I sites which showed greater change in CTCF binding in NUP153 KD cells compared to control cells and were enriched for NUP153.

14) In the list of genes differentially regulated in NUP153-KD cells, how many are bound by NUP153 directly? In their promoter or enhancer?

Based on TSS or gene body specific binding of NUP153, we identified 397 NUP153-positive differentially regulated genes (Table S6). This constituted 56% of the total differentially regulated genes (n=711).

15) In Fig 3B:

- is there a transcription change for Rtn4rl1?

We identified Rtn4rl1 as a CTCF- and NUP153-positive gene which shows altered CTCF binding and transcriptional upregulation (see RNA-Seq tracks in Figure 3C) in NUP153 KD cells.

- it seems on the figure that cohesin sites, and their change after KD, always follow CTCF kinetics. Is it always the case?

Cohesin binding sites show ~70-80% overlap with CTCF chromatin interaction sites (Parelho et al. 2008). Recently, it has been shown that cohesin positioning to CTCF binding sites is influenced by both transcription and CTCF (Parelho et al. 2008; Busslinger et al 2017). In our study, metagene profiling provided evidence that both CTCF and cohesin binding at TSS, enhancers or TAD boundaries were altered in NUP153 deficient ES cells (Figure 2D, S3D-E). Based on these facts, we envision that the cohesin phenotype is secondary to NUP153 co-function with CTCF during regulation of chromatin structure and/or gene expression.

16) In Fig 3C, the 2 critical CTCF sites (asterisks) in the HoxA cluster are not related to a NUP153 peak. Do the authors have an hypothesis?

We utilized DamID-Seq method to map NUP153 interaction sites in mouse ES cell genome. DamID relies on NUP153-Dam fusion protein to associate with the GATC sequence that resides near a NUP153 binding site, resulting in Adenine N⁶ methylation. Furthermore, Dam-fusion proteins can interact and catalyze Adenine N⁶-methylation up to ~5 kb distance (van Steensel and Henikoff. 2000). Thus, by using this method, we can detect both transient and stable NUP153 interaction sites. Lack of an immediate NUP153 peak might be due to the limitation of the DamID protocol in identification of actual binding sites. Alternatively, CTCF and NUP153 interactions might rely on interchromosomal interaction sites whereby CTCF binding at the putative binding site might be regulated through a distal NUP153 interaction site.

17) In Fig 7A, what is the significance of the change in the distance between c-fos and the nuclear periphery?

Cumulative frequency versus normalized distance graphs for the DNA FISH data set is provided in Figure 7B and Figure S7. KS-test was applied to calculate significance of the

change in c-fos locus distance to the periphery in control cells in comparison to NUP153 KD cells. Please refer to Figure 7B for P-values.

18) In general, the results are interesting in a role of NUP153 in stabilizing PolII occupancy probably through CTCF/cohesin recruitment. However, levels/numbers of changes in the KD experiments are most of the time quite minor. However this may be explained by the partial downregulation via the knock-down. A Degron system may give more significant changes.

We agree with the Reviewer that the AID system would have been superior to the other targeting technologies including shRNA which we utilized in the current study. However, we can detect clear and consistent phenotype in transcription and POL II dynamics by knocking down NUP153 using two different shRNAs in two different cell systems (HeLa and mouse ES cells). Importantly, we can rescue the transcription phenotype by restoring NUP153 levels suggesting that it is NUP153 specific. We hope that the Reviewer agrees that the use of AID system would require repeating of all the experiments and related analyses which would be timely. We think that the AID system could be utilized for future studies.

Reviewer #3 (Remarks to the Author):

The manuscript entitled "Nucleoporin 153 links nuclear pore complex to chromatin architecture by mediating CTCF and cohesin binding at cis-regulatory elements" by Dr Yildirim and colleagues describe the interaction of Nucleoporin 153 with potential interaction partners that are part of the chromatin architecture. This reviewer is mainly assessing the proteomics part of the manuscript.

We thank the Reviewer #3 for his/her excellent feedback and insightful suggestions on the proteomics data set. We are very sorry for the fact that the Reviewer could not access the proteomics data through the provided access information. We were unable to determine so as to why such a problem occurred and why the Reviewer could not access the data. All our attempts to login were successful.

The files can be opened using textpad and the data can be analyzed using Scaffold4 software (<http://www.proteomesoftware.com/products/scaffold/>). Please note that .dat files are flat text files which contain m/z peak lists of precursor ions and products. They are really only useful when importing to a database search algorithm.

We revised the manuscript according to the Reviewer's comments and recommendations. Please find our response below following the Reviewer's boldfaced comments.

Comments:

The authors are not describing the pull down experiment that well in the manuscript, especially not the database identification and quantitation. It should be described so that it could be repeated by others. As far as I understand the authors are linking the NUP153 to a FLAG tag and then use anti-FLAG affinity beads for the pull down. The control is just the beads themselves. Should the control not have been the beads with FLAG tag on, as proteins will bind to the FLAG tag? The eluted proteins are then separated on a gel and in-gel digestion is performed. The peptides are analyzed by LC-MSMS using a Q-exactive plus instrument. The database searching was performed in PD, but there is no information on how the authors have identified the true interaction partners?

1. We provide a detailed description of the IP and LC-MS/MS Proteomics Analysis below and revised the methods section accordingly. Indeed, we used a FLAG-tagged NUP153 and used anti-FLAG M2 affinity gel beads to the pull down as indicated below. We used a mock transfected cell as a control. This experimental design has been widely and successfully implemented by several groups. Based on our silver staining (Figure 1A), we had a clean background in the mock IP suggesting the specificity of our pulldown.

Immunoprecipitation (IP) assay: For IP assay, HEK293T cells that were transfected with FLAG-GFP or FLAG-NUP153 expression vector were lysed by sonication in IP lysis buffer (20 mM Tris-HCl, pH 7.9, 150 mM NaCl, 5 mM EDTA (pH:8.0), 1% Nonident P-40, 10% glycerol, 1 mM phenylmethylsulfonyl fluoride (PMSF), 1 mM DTT, protease inhibitor cocktail (Sigma-Aldrich)). After centrifugation, the supernatant was incubated with anti-FLAG M2 Affinity Gel beads (Sigma-Aldrich) at 4°C for 2 hr and the immune precipitates were subjected to western blotting. To prepare samples for the LC-MS/MS proteomics analysis, FLAG-NUP153 expression vector and mock transfected cells were lysed in elution buffer (10 mM PIPES, pH 6.8, 100 mM NaCl, 3 mM MgCl₂, 0.3 M sucrose, 0.5% Triton X-100, 1 mM PMSF, 1 mM DTT, protease inhibitor cocktail (Sigma-Aldrich)) for 10 min on ice, the nuclear fraction containing pellet was collected by centrifugation 3 min, 500 x g, 4°C and was subjected to IP assay as described above. The immune precipitates were eluted by incubation with FLAG peptide (F4799) (Sigma-Aldrich) at room

temperature for 15 min and were subjected to silver staining by using SilverXpress (Invitrogen) or utilized for LC-MS/MS proteomics analysis.

LC-MS/MS Proteomics Analysis

Samples in 1X Laemmli Sample buffer (BIO-RAD, 1610737) were run on a NuPAGE 4-12% Bis-Tris Protein gel (Invitrogen, NP0336PK2) in NuPAGE MES SDS Running Buffer (Invitrogen, NP0002) for ~5 min. The entire molecular weight range was excised and subjected to standardized in-gel trypsin digestion (<http://www.genome.duke.edu/cores/proteomics/sample-preparation/documents/In-gelDigestionProtocolrevised.pdf>). Extracted peptides were lyophilized to dryness and resuspended in 12 μ L of sample buffer (0.2% formic acid, 2% acetonitrile). Each sample was subjected to chromatographic separation on a nanoACQUITY UPLC (Waters) equipped with an ACQUITY UPLC BEH130 C₁₈ 1.7 μ m 75 μ m I.D. X 250 mm column (Waters). The mobile phase consisted of (A) 0.1% formic acid in water and (B) 0.1% formic acid in acetonitrile. Following a 3 μ L injection, peptides were trapped for 3 min on an ACQUITY UPLC M-Class Symmetry C₁₈ Trap Column 5 μ m 180 μ m I.D. X 20 mm (Waters) at 5 μ L/min in 99.9% A. The analytical column was then switched in-line and a linear elution gradient of 5% B to 40% B was performed over 30 min at 400 nL/min. The analytical column was connected to a SilicaTip emitter (New Objective) with a 10 μ m tip orifice and coupled to a Q Exactive Plus mass spectrometer (Thermo Fisher Scientific) through an electrospray interface operating in a data-dependent mode of acquisition. The instrument was set to acquire a precursor MS scan from m/z 375-1600 at R=70,000 (target AGC 1e6, max IT 60 ms) with MS/MS spectra acquired for the ten most abundant precursor ions at R=17,500 (target ABC 5e4, max IT 60 ms). For all experiments, HCD energy settings were 27v and a 20 s dynamic exclusion was employed for previously fragmented precursor ions. Raw LC-MS/MS data files were processed in Proteome Discoverer (Thermo Fisher Scientific) and then submitted to independent Mascot search (Matrix Science) against a SwissProt database (*Human* taxonomy) containing both forward and reverse entries of each protein (20,322 forward entries). Search tolerances were 5 ppm for precursor ions and 0.02 Da for product ions using trypsin specificity with up to two missed cleavages. Carbamidomethylation (+57.0214 Da on C) was set as a fixed modification, whereas oxidation (+15.9949 Da on M) and deamidation (+0.98 Da on NQ) were considered dynamic mass modifications. All searched spectra were imported into Scaffold (v4.4, Proteome Software) and scoring thresholds were set to achieve a peptide false discovery rate of 1% using the PeptideProphet algorithm.

The database searching was performed in PD, but there is no information on how the authors have identified the true interaction partners?

As indicated in the Methods section and above, we used the following criteria to identify interacting partners.

“Raw LC-MS/MS data files were processed in Proteome Discoverer (Thermo Fisher Scientific) and then submitted to independent Mascot search (Matrix Science) against a SwissProt database (*Human* taxonomy) containing both forward and reverse entries of each protein (20,322 forward entries). Search tolerances were 5 ppm for precursor ions and 0.02 Da for product ions using trypsin specificity with up to two missed cleavages. Carbamidomethylation (+57.0214 Da on C) was set as a fixed modification, whereas oxidation (+15.9949 Da on M) and deamidation (+0.98 Da on NQ) were considered dynamic mass modifications. All searched spectra were imported into Scaffold (v4.4, Proteome Software) and scoring thresholds were set to achieve a peptide false discovery rate of 1% using the PeptideProphet algorithm.”

And according to the manuscript the experiment was performed once? IP and protein identification is associated with a lot of errors and pull down experiments should be performed at least 3 times in order to say anything quantitative about protein identifications, especially when the peptides comes from in-gel digestion.

The mass-spec dataset was intended to generate qualitative characterization of possible unique interactors of NUP153 and the experiment was carried out once. After determining NUP153 interacting proteins (i.e. cohesin subunits RAD21, SMC1 and SMC3), we further validated their interaction with NUP153 by biochemical means using co-IP experiments (Figure 1B).

This reviewer tried for several days to download the raw data files from ProteomeXchange using the Username and password provided but it was not possible. So it is not possible to evaluated the protein identifications or quantitation.

We are very sorry that the Reviewer could not access the proteomics data using the provided access information. We were unable to determine so as to why such a problem

occurred and why Reviewer #3 could not access the data. All our attempts to login were successful.

Proteomics data have been deposited to the ProteomeXchange Consortium via the PRIDE partner repository with Project ID: PXD01544.

Access link: <https://www.ebi.ac.uk/pride/archive/login>

Username: reviewer78224@ebi.ac.uk

Password: provided to the Editor.

The files can be opened using textpad and the data can be analyzed using Scaffold4 software (<http://www.proteomesoftware.com/products/scaffold/>). Please note that .dat files are flat text files which contain m/z peak lists of precursor ions and products. They are only useful when importing to a database search algorithm.

REVIEWERS' COMMENTS:

Reviewer #1 (Remarks to the Author):

In the provided revision, the authors have re-worked their data analysis to address my concerns, and the presentation of the data in Figures 2 and 3 (and S2, S3) is now much more clear and convincing, with a stronger message. They have also added other controls and clarifications, greatly improving the manuscript. I agree with the authors that it is not necessary to include additional 3C/4C data in the present manuscript.

Reviewer #2 (Remarks to the Author):

The authors provided a revised version of the manuscript aiming to emphasize how the NUP153 factor links nuclear pore complexes and chromatin architecture, through the genome organizers CTCF and cohesin, to regulate gene expression. The manuscript has been improved and answers to most of the reviewers comments, to my opinion.

In particular:

1. They included controls, in particular for FLAG-NUP153 expression, and comments when necessary.
2. They included data and comments on the association of NUP153 to TAD boundaries that strengthen findings for a role of NUP153 on chromatin architecture.
3. They revised analyses of genome-wide analyses in mouse ES cells, in particular NUP153 binding on TSS, enhancers and TAD boundaries.
4. They simplified grouping of CTCF-binding sites in 2 groups: group1 with greater binding in WT versus NUP153-KD, and group2 with similar or weaker binding.
5. They aimed to better connect the two parts of the manuscript.

Reviewer #3 (Remarks to the Author):

The Authors have addressed all this reviewers concerns. This reviewer has downloaded the raw data files from the proteomics pull down experiment and the analysis look ok. The validation of some targets with WB should be fine.

April 19, 2020

POINT-BY-POINT RESPONSE TO REVIEWERS

We thank all three Reviewers for their time, consideration and valuable feedback on our revised manuscript entitled "*Nucleoporin 153 links nuclear pore complex to chromatin architecture by mediating CTCF and cohesin binding at cis-regulatory elements and TAD boundaries*" (NCOMMS-19-31033A). As stated in the bold-faced Reviewers' Comments below, the Reviewers did not provide any additional concerns or comments to be addressed. We were very pleased to learn that the Reviewers indicated that we addressed all their concerns and comments. We were also very glad to hear that the Reviewers found the revised version of our manuscript greatly improved, clearer and more convincing with a stronger message.

REVIEWERS' COMMENTS:

Reviewer #1 (Remarks to the Author):

In the provided revision, the authors have re-worked their data analysis to address my concerns, and the presentation of the data in Figures 2 and 3 (and S2, S3) is now much more clear and convincing, with a stronger message. They have also added other controls and clarifications, greatly improving the manuscript. I agree with the authors that it is not necessary to include additional 3C/4C data in the present manuscript.

Reviewer #2 (Remarks to the Author):

The authors provided a revised version of the manuscript aiming to emphasize how the NUP153 factor links nuclear pore complexes and chromatin architecture, through the genome organizers CTCF and cohesin, to regulate gene expression. The manuscript has been improved and answers to most of the reviewers comments, to my opinion.

In particular:

1. They included controls, in particular for FLAG-NUP153 expression, and

comments when necessary.

2. They included data and comments on the association of NUP153 to TAD boundaries that strengthen findings for a role of NUP153 on chromatin architecture.
3. They revised analyses of genome-wide analyses in mouse ES cells, in particular NUP153 binding on TSS, enhancers and TAD boundaries.
4. They simplified grouping of CTCF-binding sites in 2 groups: group1 with greater binding in WT versus NUP153-KD, and group2 with similar or weaker binding.
5. They aimed to better connect the two parts of the manuscript.

Reviewer #3 (Remarks to the Author):

The Authors have addressed all this reviewers concerns. This reviewer has downloaded the raw data files from the proteomics pull down experiment and the analysis look ok. The validation of some targets with WB should be fine.